

# The Atmospheric Composition Component of the ICON modeling framework: ICON-ART version 2025.04

Gholam Ali Hoshyaripour[1], Andreas Baer[1], Sascha Bierbauer[1], Julia Bruckert[1], Dominik Brunner[3], Jochen Förstner[2], Arash Hamzehloo[3], Valentin Hanft[1], Corina Keller[3], Martina Klose[1], Pankaj Kumar[1], Patrick Ludwig[1], Enrico Metzner[1], Lisa Muth[1], Andreas Pauling[4], Nikolas Porz[2], Thomas Reddmann[1], Luca Reißig[1], Roland Ruhnke[1], Khompat Satitkovitchai[1], Axel Seifert[2], Miriam Sinnhuber[1], Michael Steiner[3], Stefan Versick[1], Heike Vogel[1], Michael Weimer[1,5], Sven Werchner[1], and Corinna Hoose[1]

[1]Karlsruhe Institute of Technology (KIT), Karlsruhe, Germany
[2]Deutscher Wetterdienst (DWD), Offenbach am Main, Germany
[3]Swiss Federal Laboratories for Materials Science and Technology (EMPA), Dübendorf, Switzerland
[4]Federal Office of Meteorology and Climatology MeteoSwiss, Zurich, Switzerland
[5]Institute of Environmental Physics (IUP), University of Bremen, Bremen, Germany

**Correspondence:** Gholam Ali Hoshyaripour (ali.hoshyaripour@kit.edu)

**Abstract.** Accurate and efficient modeling of atmospheric composition, including aerosols and trace gases and their interactions with radiation, clouds, and dynamics is essential for improving predictions of air quality, weather, climate, and related health impacts. The ART (Aerosols and Reactive Trace gases) component extends the ICOsahedral Nonhydrostatic (ICON) modeling framework by enabling online, fully coupled simulations of atmospheric composition processes across scales. ART

includes modules for emissions, transport, gas-phase chemistry, and aerosol microphysics in both the troposphere and stratosphere, allowing for the investigation of feedbacks between atmospheric composition and physical processes from the large-eddy to global scale.

This paper presents an updated overview of the ICON-ART framework as implemented in version 2025.04, highlighting recent developments in emission parameterizations, chemical mechanisms, aerosol processes, and coupling to the physical

core of ICON via aerosol–radiation and aerosol–cloud interactions. We summarize the structure of the code infrastructure and demonstrate the model's flexibility and scalability across a wide range of applications. ICON-ART provides a unified and modular platform for research and operational use in atmospheric composition, bridging the gap between regional air quality modeling and global Earth system simulations.

## 1   Introduction

Atmospheric composition focuses on the variations in and processes affecting trace gases and aerosols, which in turn influence air quality, weather, and climate. These components and their interactions with radiation and clouds are vital for processes such as cloud formation, radiative forcing, and precipitation patterns. Therefore, accurately simulating atmospheric composition is



essential for improving predictions related to weather, renewable energy, climate change, air pollution, and associated health impacts.

The mutual feedback between the chemical and physical states of the atmosphere across different scales has driven the integration of atmospheric composition modeling into weather and climate models (Baklanov, 2010; Grell and Baklanov, 2011; Brasseur and Kumar, 2021). Beginning in the early 2010s, several global and regional-scale model systems have been developed to account for the interactions between atmospheric composition and the physical state of the atmosphere (Baklanov et al., 2014). This led to the development of online-coupled models ranging from global hydrostatic chemistry-climate models

like ECHAM/HAMMOZ (Schultz et al., 2018), EMAC (Jöckel et al., 2006), and WACCM-X (Liu et al., 2018) to regional non-hydrostatic models, such as WRF-Chem (Grell et al., 2005) and COSMO-ART (Vogel et al., 2009). One particular focus of these developments has been on how aerosols, trace gases, and their interactions with radiation and clouds influence weather patterns, air quality, and climate. Recent advancements in modeling aerosol-radiation interaction (ARI) (Rieger et al., 2017; Hoshyaripour et al., 2019; Yang et al., 2020; Oh et al., 2024), aerosol-cloud interaction (ACI) (Glotfelty et al., 2019; Zhang

et al., 2022; Seifert et al., 2023; Samanta et al., 2024), and the integration of chemistry into weather forecasting models (Kukkonen et al., 2012; Hodzic and Madronich, 2018; Deroubaix et al., 2024) have significantly improved our ability to predict these complex systems. However, only a few models can account for local, regional, and global weather and climate processes within a single modeling framework.

The ICOsahedral Nonhydrostatic weather and climate model (ICON) has been developed and widely used for weather and

climate prediction across scales. It solves the 3D non-hydrostatic and compressible Navier–Stokes equations on an icosahedral-triangular grid (Gassmann and Herzog, 2008), facilitating precise predictions across scales (Zängl et al., 2015; Heinze et al., 2017; Giorgetta et al., 2018). The module Aerosols and Reactive Trace gases (ART), integrated into the ICON framework, enables comprehensive modeling of atmospheric composition. It handles emissions, transport, and transformations of trace gases and aerosols, incorporating gas-phase chemistry and aerosol dynamics in the troposphere and stratosphere (Rieger et al.,

2015; Weimer et al., 2017; Schröter et al., 2018). ICON-ART has been successfully used to investigate mutual feedbacks between the chemical and physical states of the atmosphere across different scales ranging from large-eddy simulations (Muth et al., 2025) to regional weather (Rieger et al., 2017; Seifert et al., 2023) and global climate (Weimer et al., 2021).

Previous works have described ICON-ART with respect to the basic equations, parameterizations and numerical methods (Rieger et al., 2015), tracer framework (Schröter et al., 2018) and chemistry processes (Weimer et al., 2017; Schröter et al.,

2018). This paper provides an updated overview of the ICON-ART framework version 2025.04, highlighting the recently developed components and features that enable its role in atmospheric composition modeling. We discuss the emission parameterizations in section 2, followed by the description of chemistry and aerosol processes in sections 3 and 4, respectively. Then we explain interactions within the broader ICON modeling framework through aerosol-radiation and aerosol-cloud interactions. A summary of the code infrastructure is given in section 6 followed by conclusions and outlook in section 7.





## 2 Emission processes

ART accounts for the emission of gases and particulate matter from both natural and anthropogenic sources, as listed in Table 1. The emission processes for wildfires, desert dust, sea salt, dimethyl sulfide (DMS), pollen, and the online emission module (OEM) are taken from the original implementations in the COSMO-ART framework. Other emission parameterizations have been developed and implemented within the ICON-ART framework. In both cases, significant modifications have been made

for scientific or technical reasons, which are detailed below. For further information on the parameterizations, see the references in Table 1.

Table 1: Emissions in the ICON-ART model system.

| Emission Type | Basis | Implementation in ART |
|---|---|---|
| Anthropogenic | Prescribed (Weimer et al., 2017), OEM (Jähn et al., 2020) | Weimer et al. (2017); Jähn et al. (2020), see Sect. 2.1 |
| Wildfires | GFAS (Kaiser et al., 2012) and Plume-rise model (Freitas et al., 2007) | Walter et al. (2016), see Sect. 2.2 |
| Volcanic | 1D model FPlume (Folch et al., 2016) | Bruckert et al. (2022), see Sect. 2.3 |
| Desert Dust | Saltation-based (Vogel et al., 2006) | Rieger et al. (2017) |
| Sea Salt | Wave breaking and whitecap formation (Monahan et al., 1986; Smith and Harrison, 1998; Mårtensson et al., 2003; Grythe et al., 2014) | Lundgren et al. (2013); Rieger et al. (2015); see Sect. 2.4 |
| DMS | DMS conc. in ocean (Lana et al., 2011) | see Sect. 2.5 |
| Biogenic VOCs | MEGAN (Guenther et al., 2012) | Weimer et al. (2017) |
| Pollen | EMPOL (Zink et al., 2013) | Zink et al. (2013), see Sect. 2.6 |
| Point source | Rieger et al. (2015) | Rieger et al. (2015) |

### 2.1 Online Emission module

The Online Emission Module (OEM) was first developed for COSMO-ART (Jähn et al., 2020) and then adapted to ICON-ART and refactored for improved computational performance. OEM enables efficient processing of emissions that are constant

in time or changing only temporally, but not spatially. This holds for most applications requiring input from anthropogenic emission inventories such as the Emission Database for Global Atmospheric Research (EDGAR) (Crippa et al., 2018) or the regional inventory CAMS-REG (Kuenen et al., 2022). These inventories provide gridded emissions divided into individual source categories. An example for such a source classification is the Gridded Nomenclature For Reporting (GNFR), which





distinguishes 12 source categories from public power to agricultural emissions (Super et al., 2020). The main steps for inte-
grating these inventories into the model are (i) re-mapping to the ICON grid, (ii) application of temporal scaling factors, (iii)
vertical distribution of the emissions onto ICON's terrain-following hybrid sigma-z coordinates, and (iv) summing up over
all source categories. In most model systems including WRF-Chem, CHIMERE or CAMx, [e.g.,][](Menut et al., 2024; Woo
et al., 2012), these steps are performed externally using a pre-processing software such as Hermes (Guevara et al., 2019), which
generates a large number of files each containing the emission field representative of a given time interval (e.g. hourly), which
are then read into the model during runtime. OEM, instead, allows reading the emissions together with all temporal and vertical
scaling information only once during model initialization. The temporal and vertical scalings are then applied online during
the simulation. This greatly reduces the need for data pre-processing and simplifies the setup of new simulations. As shown in
Jähn et al. (2020), the additional time required for the online computations is fully compensated by the time saved through less
frequent file access.

Inputs for OEM can be produced with the Python package emiproc (Constantin et al., 2025), which is able to process multiple
emission inventories to prepare the inputs for a range of atmospheric transport models including ICON-ART. The tool maps the
emissions in a mass-conserving way onto the ICON grid and creates all additional inputs including source and country-specific
temporal and vertical profiles and the corresponding country masks. The tool is also able to merge multiple inventories, for
example to embed a high-resolution national inventory into a coarser global inventory.

To compute the emission of a species $X$ at time $t$ in a given grid cell, OEM performs the following operation

$$E_X(z,t) = \sum_{s=0}^{N_s} E_{X,s} \cdot w_{X,s}(t) \cdot v_{X,s}(z), \tag{1}$$

with $E_{X,s}$ the emission of species $X$ from source category $s$ in that grid cell from the inventory, $w_{X,s}(t)$ the temporal scaling
factor at time $t$ for source category $s$, and $v_{X,s}(z)$ the vertical scaling factor for source category $s$ at vertical level $z$. $N_s$ is the
total number of categories. The temporal factor $w_{X,s}(t)$ is computed as the product of three different scaling factors describing
diurnal, day-of-week, and seasonal variability

$$w_{X,s}(t) = w_{X,s,h}(h(t)) \cdot w_{X,s,d}(d(t)) \cdot w_{X,s,m}(m(t)) \tag{2}$$

with $h(t)$ being the hour of the day, $d(t)$ the day of the week, and $m(t)$ the month of the year. Alternatively, a separate scaling
factor can be defined for each hour of the year to represent, for example, heating emissions varying with outdoor temperatures.
Furthermore, different temporal scaling factors can be provided for different countries (or regions) together with a country
mask. Diurnal factors $w_{X,s,h}(h(t))$ are computed with respect to local time.

In addition to temporal and vertical factors, it may be necessary to provide speciation factors, which describe the fractional
contribution of individual model species $X$ to the total emissions of a family of species $\tilde{X}$ in the inventory. Examples are
non-methane volatile organic compounds (NMVOC) and nitrogen oxides ($NO_x$), for which inventories typically provide only
the total emissions of the family but not of the individual compounds. Speciation factors cannot be supplied as input to OEM,





but are dealt with by emiproc. Based on an emission field for the family $\tilde{X}$ and a set of speciation factors, emiproc generates and writes out emission fields for all model species $X$.

## 2.2 Wildfires

The ICON-ART model system incorporates a one-dimensional, sub-grid-scale plume-rise model developed by Freitas et al. (2006, 2007, 2010), similar to its implementation in COSMO-ART by Walter et al. (2016). This plume-rise model is suited for applications with horizontal resolutions on the order of 10 to 100 km. For these applications, the model calculates plume height based on buoyancy, atmospheric stratification, and flow conditions, accounting for processes that occur on scales much smaller than the horizontal spacing of the ICON. The model uses an internal vertical grid spacing of 100 meters with 200 vertical layers. Environmental conditions (pressure, humidity, temperature, wind speed) are provided by ICON and transferred to the plume-rise model as initial and environmental conditions for each active fire grid point to determine plume height.

Fire size and intensity, based on the ICON land use class, vegetation type and density, determine heat release and initial buoyancy. The lower boundary condition assumes a virtual buoyancy source below the surface, resulting in high vertical velocity. Final buoyancy is limited by turbulent and dynamic entrainment, with additional buoyancy from latent heat release during condensation. The plume top is defined where vertical velocity inside the plume drops below $1\,\mathrm{m\,s^{-1}}$ after an equilibrium state between the surroundings and the heat source is achieved. Heat flux values, dependent on vegetation type, are taken from Freitas et al. (2006). They were corrected in comparison to the implementation in COSMO-ART (Walter et al., 2016) such that the plume-rise model exactly reproduces the results in Freitas et al. (2010).

In addition to the original numerical solver introduced by Freitas et al. (2007), a more efficient and stable first-order implicit solver for the same equations was developed and implemented. The solver relies on a Godunov type scheme to solve the inviscid Burgers equation for the vertical velocity inside the plume and on upwind schemes for the other modeled plume variables such as temperature, specific humidity, specific cloud water, specific rain water, specific ice, horizontal entrainment velocity, and plume radius. An internal grid is not needed for this solver since it copes with unevenly spaced grids like the vertical atmosphere discretization in numerical weather forecast models such as ICON while at the same time allowing much longer plume internal time-steps. The solver comes with an a priori condition that determines whether a height calculation is needed or if the plume stays at the minimal emission height. The plume heights strongly vary with the time of day since a diurnal cycle function $d$ is applied to fire intensity and size. The function proposed by Kaiser et al. (2009); Andela et al. (2015) and applied by Walter et al. (2016) in COSMO-ART is also used in ICON-ART. The plume-rise model returns plume bottom and top heights, with a parabolic emission profile $f$ describing the vertical distribution of emissions. The emission rate $E$ of a species in $\mathrm{kg\,m^{-2}\,s^{-1}}$ is calculated for each grid cell based on height $z$ and time $t$.

$$E_X(z,t) = M_X(t) \cdot d(t) \cdot f(z,t) \tag{3}$$

$M_X$ is the daily mean emission flux of species $X$ from CAMS GFAS (Kaiser et al., 2012), $d$ is the diurnal cycle, and $f$ is the parabolic emission profile between upper and lower injection heights.



CAMS GFAS relies on Fire Radiative Power (FRP) from the NASA MOD14 product, which includes thermal radiation observations from the MODIS instrument (Kaiser et al., 2012). To address data gaps due to, e.g. cloud cover, fire data is assimilated using a Kalman filter and statistics. FRP density is updated based on previous and current observations, with sampling limited to four times per day to represent the diurnal fire cycle (Kaiser et al., 2012). GFAS data is provided at a resolution of 0.1°.

## 2.3 Volcanic eruptions

The rise of volcanic plumes during eruptions depends on both volcanic and atmospheric conditions. Volcanic conditions are the exit temperature, exit velocity, exit volatile fraction, and the vent diameter. The exit velocity and the vent diameter control the mass eruption rate (MER). The height of the plume depends on volcanic conditions due to effects on the MER as well as on the plume density and atmospheric conditions. Due to the complexity of plume dynamics, simple relationships (e.g., Mastin et al., 2009), which only depend on plume height and are often used in dispersion models, can lead to large uncertainties in emissions (e.g., Marti et al., 2017; Bruckert et al., 2022).

Volcanic emissions in ICON-ART are calculated online using the 1-D volcanic plume model FPlume by Folch et al. (2016), which considers the volcanic conditions as well as processes during the plume rise such as ambient air entrainment, plume bending due to wind, particle wet aggregation, energy supply due to water phase changes, and particle fallout and re-entrainment. Bruckert et al. (2022) described the coupling of FPlume with ICON-ART in detail. In short, FPlume requires atmospheric profiles for temperature, pressure, density, zonal and meridional wind speed, and specific humidity at the volcanic vent. In addition to meteorological data, FPlume needs estimates of the exit temperature, exit velocity, and exit volatile fraction, which depend on the type and setting of the volcano. FPlume can either calculate the MER based on a given height or the height based on a given MER. During every time step when the volcano is active, FPlume first calculates the plume properties, i.e., the total MER in the case of a given plume height or plume height in the case of a given MER. Second, the fraction of very fine ash, which is relevant for long-range transport, is determined based on the plume height and the total MER by using the relationship of Gouhier et al. (2019). Third, very fine ash is emitted along a profile that has initially been defined by Suzuki (1983) and applied by Marti et al. (2017) for the coupling of FPlume to NMMB-MONARCH-ASH transport model (Nonhydrostatic Multiscale Model on the B-grid – Multiscale Online Nonhydrostatic AtmospheRe CHemistry model – ASH). The distribution of the very fine ash mass into the ICON-ART modes is prescribed in the FPlume input file.

$SO_2$ can be emitted alongside with ash using the same timing and profile, however, the MER of $SO_2$ is prescribed in the FPlume input file, as FPlume does not differentiate between gaseous compounds and water vapor. Bruckert et al. (2025) extended the coupling of ICON-ART and FPlume by water vapor emission. Here, the MER of water vapor is calculated from the exit water mass fraction and the total MER and is emitted through the same profile as ash. This simplification neglects the entrainment of water vapor and was so far only tested for the water-rich 2022 Hunga eruption (Bruckert et al., 2025).

The advantages of the coupling are (1) a more accurate MER and therefore better agreement with observed mass column loadings as shown for the 2019 Raikoke eruption, Kuril Islands (Bruckert et al., 2022) and (2) an easy consideration of eruption





phases, which allows a comparison to observations in the near-field of complex volcanic emissions such as the 2021 La Soufrière eruption, St. Vincent (Bruckert et al., 2023).

## 2.4 Sea salt

For the emission of sea salt, ICON-ART offers two options. The first option is based on the parameterizations of Monahan et al. (1986), Mårtensson et al. (2003), and Smith and Harrison (1998) as described by Rieger et al. (2015). Another option is

based on Grythe et al. (2014) which is described below.

Numerous studies have shown that sea surface temperature (SST) significantly influences the emission rates of sea salt by affecting the physical properties of foam and droplet formation at the ocean surface. To develop a globally applicable source function that can also represent realistic sea salt concentrations in tropical regions, incorporating this temperature dependence is essential (Grythe et al., 2014). To address this, we implemented an alternative scheme for sea salt emission in ICON-ART

based on Grythe et al. (2014), which in addition to wind speed includes SST as a key parameter:

$$
\begin{aligned}
\frac{dF(D_p, U_{10}, T)}{dD_p} = T_w \cdot \Bigg( & 235 \cdot U_{10}^{3.5} \exp\left(-0.55 \left(\ln\left(\frac{D_p}{0.1}\right)\right)^2\right) + 0.2 \cdot U_{10}^{3.5} \exp\left(-1.5 \left(\ln\left(\frac{D_p}{3}\right)\right)^2\right) \\
& + 6.8 \times 10^{-3} \cdot U_{10}^{3} \exp\left(-1 \left(\ln\left(\frac{D_p}{30}\right)\right)^2\right) \Bigg).
\end{aligned}
\tag{4}
$$

In this equation, $F(D_p, U_{10}, T)$ is the sea salt emission flux in $\mathrm{particles\,m^{-2}\,s^{-1}}$, $D_p$ is the dry particle diameter in $\mu$m, $U_{10}$ is the 10-meter wind speed in m s$^{-1}$, $T$ is the SST in K, and $T_w$ is an empirical temperature correction factor (dimensionless), accounting for the effect of SST on sea salt emissions. This parameterization is applied to three sea salt modes with median di-

ameters of 0.1, 3.0 and 30 $\mu$m, with standard deviations of 1.9, 2.0 and 1.7, respectively (Grythe et al., 2014). This enhancement enables a more accurate representation of the regional and seasonal variability of sea salt concentrations, especially in tropical and subtropical ocean regions where conventional parameterizations often underestimate emissions. The new parameterization thus represents an important step towards a more physically consistent modeling of marine aerosol sources on a global scale. We note that the factor $10^{-3}$ in the last term of the equation is mistakenly missing in the original publication by Grythe et al.

(2014) and is corrected here.

## 2.5 DMS

DMS has multiple sources with one large contribution from oceanic biogenic emissions (Lana et al., 2011). As such, these emissions are highly dependent on the exchange between ocean and atmosphere and hence on wind speed. Therefore, an online calculation is needed to account for DMS emissions from the ocean.

Lana et al. (2011) provided a 1°x1° monthly climatology of DMS ocean surface concentrations ($C_w$) based on measured data. These climatological values are converted in ICON-ART to the DMS emission flux $F_{\mathrm{DMS}}$ using the following parameterization (Lundgren et al., 2013; Ullwer, 2018):





$$F_{\mathrm{DMS}} = \frac{C_{\mathrm{w}} \cdot M_{\mathrm{DMS}}}{3.6 \cdot 10^2} \left( 0.222\, u_{10\,\mathrm{m}}^2 + 0.333 u_{10\,\mathrm{m}} \right). \tag{5}$$

In this equation, $M_{\mathrm{DMS}} = 6.21\,\mathrm{g\,mol^{-1}}$ is the DMS molar weight, $C_{\mathrm{w}}$ has the unit $\mathrm{mol\,l^{-1}}$ and $u_{10\,\mathrm{m}}$ is the wind speed

at the altitude of $10\,\mathrm{m}$ above sea level in $\mathrm{m\,s^{-1}}$. The emission flux is added to the DMS tracer mixing ratio using the same procedure as for other emissions (Weimer et al., 2017).

### 2.6 Pollen

The pollen emission model is based on the EMPOL approach (Zink et al., 2013). The basic idea is that the emission process can be divided into two steps: (1) opening of the anthers and the accumulation of pollen in a reservoir; (2) release of pollen

from the reservoir into the air, driven by turbulence.

The opening of the anthers is controlled by phenology, temperature and humidity. The phenology is modeled using a temperature sum approach, as described by Pauling et al. (2014). In general, warm and dry conditions favor opening of the anthers. To calculate the amount of pollen released into the reservoir, a plant distribution map is required. When turbulence is sufficiently strong, the pollen reservoir is emptied and the pollen becomes airborne. This process is parameterized using the Turbulent

Kinetic Energy (TKE). Once airborne, pollen is treated as a passive tracer and can be removed by dry and wet deposition. Re-suspension of pollen is not considered by the model.

The pollen emission model was originally designed for COSMO-ART and later implemented in ICON-ART. Currently, implemented plant species include hazel, alder, birch, grasses and ambrosia. Each species is handled slightly differently within the model. Recent developments include an emission implementation for hazel and the integration of real-time pollen data.

The approach is described by Adamov and Pauling (2023). The real-time pollen data is used in two ways. First, the start of the flowering season is set to the date when the first pollen are observed. Second, the emission flux is scaled so that the modeled concentrations match the observed values.

## 3   Chemistry processes

In ART different types of tracers and solvers for them can be chosen. Passive tracers are inert and are only changed by

emission and transport processes. Region tracers are a specific subset of passive tracers that originate from predefined regions. Chemtracers are tracers that have a simplified solver scheme, e.g. Linoz for ozone. Meccatracers are the most complex tracers in ICON and are solved by an Ordinary Differential Equation (ODE) scheme. An overview is given in Table 2.

Table 2: Types of chemistry in ART

| Chemistry Type | Basis | Implementation in ART |
|---|---|---|
| Region tracers | Vogel et al. (2015) | Rieger et al. (2015) |





| | | |
|---|---|---|
| Passive tracers | Rieger et al. (2015) | Rieger et al. (2015) |
| Lifetime | Rieger et al. (2015) | Rieger et al. (2015) |
| Linoz v2 | Mclinden et al. (2000) | Schröter et al. (2018) |
| Linoz v3 | Hsu and Prather (2010) | Ramezani Ziarani et al. (2025) |
| SimNOY | Diekmann (2021) | Diekmann (2021) |
| UBCNOy | Funke et al. (2016); Matthes et al. (2017) | Ramezani Ziarani et al. (2025) |
| OH chemistry | Weimer et al. (2017) | Weimer et al. (2017) |
| MECCA | Sander et al. (2019) | Schröter et al. (2018) |

## 3.1 Simplified chemistry options

### 3.1.1 Region tracers

Understanding the origin and subsequent pathways of air masses is fundamental to interpreting atmospheric composition and evaluating the transport characteristics of atmospheric models. To facilitate these investigations within the ICON-ART modeling system, a suite of passive "region tracers" has been incorporated.

     These tracers serve as computationally efficient diagnostic tools, each representing a distinct geographical source region. The defined regions encompass a variety of scales and types, including continental regions (e.g., Europe, North America, East

Asia), oceanic basins (e.g., Tropical Pacific, Tropical Atlantic) and broad hemispheric backgrounds. Besides region tracers described in Vogel et al. (2015), additional tracers used in the PHILEAS campaign and ASCCI campaign are implemented. Those tracers are set to 1 inside and 0 outside their source region at the lowest level within ICON-ART.

     Conceptually, air originating within the lowest model layer of a specific source region is "tagged" with its corresponding tracer. Those tracers are then transported throughout the model domain solely by the simulated atmospheric dynamics. These

tracers are inert; they do not undergo chemical transformation or deposition processes. As a result, the concentration of a specific region tracer at any location and time within the simulation directly quantifies the fractional contribution of air that originated from that source region. In contrast to Vogel et al. (2015) in ICON-ART a land-sea mask is added to the tracers.

     An example usage of those tracers used during the ASCCI campaign is shown in Fig. 1. On the left, the source regions of the tracers are displayed. On the right, the distribution at 300 hPa of the region tracer originating from Europe is shown for

March, 23rd 2025. The simulation was started on December 1st, 2024 and reinitialized each day with the current meteorological conditions provided by DWD. In the figure a distinct narrow filament can be seen over the north-west of Scandinavia. The blue lines with red dots depict the corresponding flight path of the research aircraft HALO, where in-situ and remote sampling of the structure has been conducted.





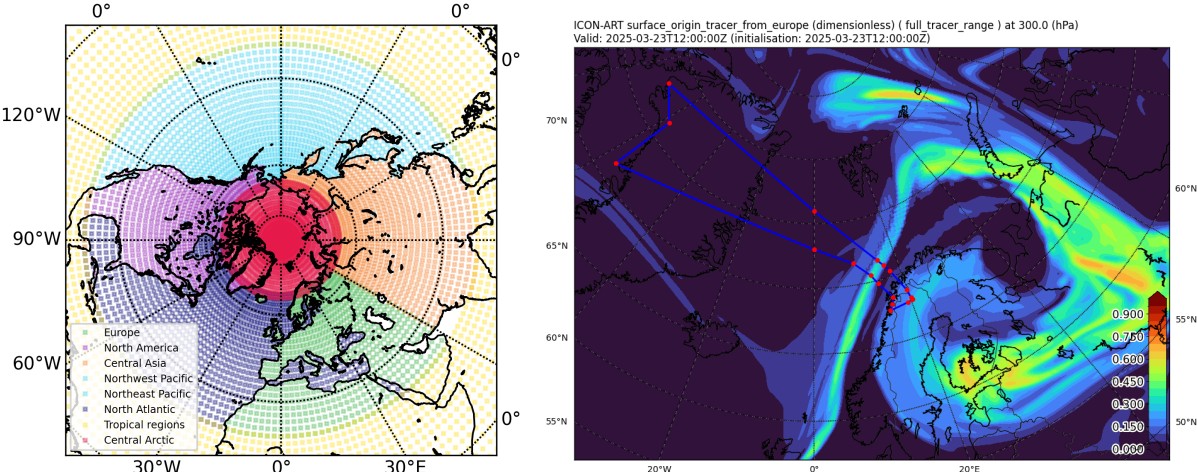

**Figure 1.** Left: source regions for region tracers used during the ASCCI airplane measurement campaign; Right: Distribution of tracer originating from Europe in 300 hPa, as well as the charted flight path of Flight 09 of the ASCCI campaign.

### 3.1.2 Lifetime tracers

Lifetime tracers in ICON-ART should be used when a very fast chemistry, but not the highest accuracy is needed. Some tracers in ICON-ART get a special treatment when they have specific names and when "lifetime" is chosen as solver. Tracers with lifetimes changing vertically as a function of pressure or overlaying $O_2$ column are those presented in Fig. 2. The corresponding formula are summarized in Appendix F. For other tracers with "lifetime´´ chosen as solver, the globally constant lifetime given in an XML file is used.

Combined effects on TRCO2 compared to a tracer named CO2 of the lifetime approach and $CO_2$ deposition in the ocean can be found Fig. 7.

### 3.1.3 Linoz v2: simple atmospheric ozone scheme

One way to calculate ozone concentrations in ICON-ART is the implementation of the LINearized OZone scheme as described by Mclinden et al. (2000). It is a fast online calculation of ozone as a chemical tracer using a linear approximation for the ozone change depending on temperature, overhead ozone column, local mixing ratio, and optionally, as described in more detail below, polar depletion. An in-depth explanation of the implementation in ICON-ART can be found in Schröter et al. (2018). Since the publishing of the aforementioned paper, several bug fixes have been applied to LINOZ as well as an adjustment to include the near-surface relaxation to a globally constant ozone volume mixing ratio of 25 ppb, as proposed already by Mclinden et al. (2000). Comparing the same setup (40 km horizontal resolution, 90 height levels, 48 hours lead time, 10 test runs) with and without LINOZ shows an increase in computational runtime for the entire model run of up to 7%.





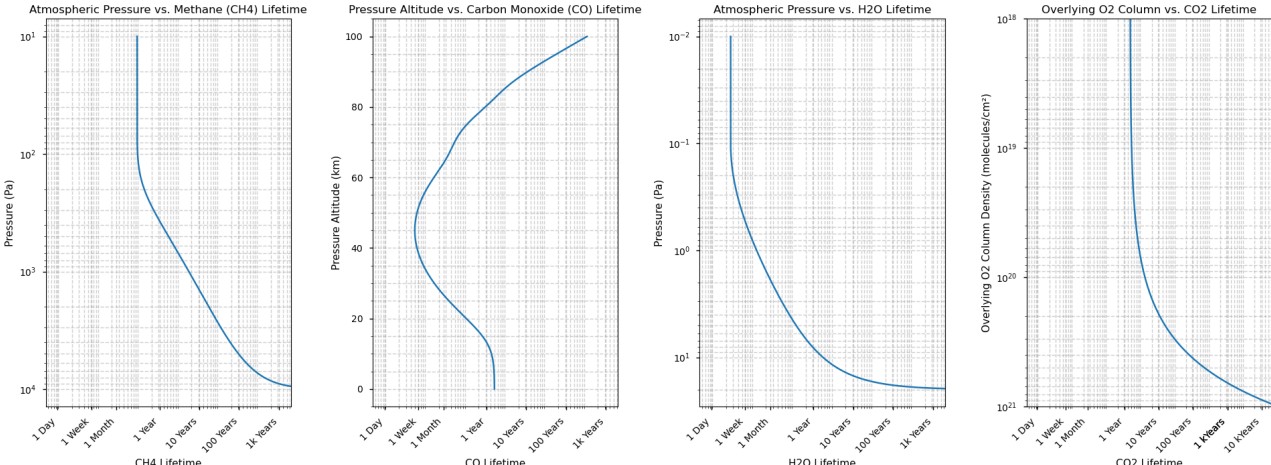

**Figure 2.** Lifetimes from left to right for the tracers TRCH4, TRCO, TRH2O and TRCO2

LINOZ provides the possibility to include an additional parametrization for polar ozone loss due to catalytic ozone depletion by chlorine and bromine. This parametrization is activated in model cells that are located in polar regions ($|lat| > 45°$), have a temperature below 195 K and a solar zenith angle smaller than a fixed value of 85° (cold tracer parametrization) or 90° (lifetime parametrization). The cold tracer parametrization contains an additional tracer that emulates the prolonged activation time of these species after their initial activation. Figure 3 shows ozone columns when using LINOZ with polar loss parametrization and coldtracer activated. In addition to that, the chlorine and bromine loading used in the parametrization of LINOZ has been adapted according to Hossaini et al. (2019). Ozone is initalized with CAMS EAC4 data (Inness et al., 2019) and then runs according to the LINOZ calculation for four months with meteorological reinitialization every 24 hours.

### 3.1.4 Linoz v3 and UBCNOy: Solar forcing of stratospheric ozone

To include variable solar forcing by energetic electron precipitation (EEP) and spectral solar irradiance (SSI) via stratospheric ozone into ICON-ART, the following adaptations were made:

– **UBCNOy**: An upper boundary condition of $NO_y$ (NO, $NO_2$, $NO_3$, 2 $N_2O_5$, $HNO_3$, $HNO_4$, $ClNO_3$) was implemented at three model levels below the model top boundary. UBCNOy is based on a semi-empirical model of the auroral and magnetospheric electron precipitation into the mesosphere and lower thermosphere using the geomagnetic Ap-Index as prognostic variable and constrained by MIPAS satellite observations (Funke et al., 2016) also part of the solar forcing recommendations for chemistry-climate models, e.g., for CMIP6 and CMIP7 (Matthes et al., 2017; Funke et al., 2024). The upper boundary $NO_y$ was added to the stratospheric $NO_y$ background derived from SimNOy (see Sec. 3.1.5). The SimNOy tables were extended to the mesopause by using output from the EMAC model to account for the mesospheric lifetime of $NO_y$.



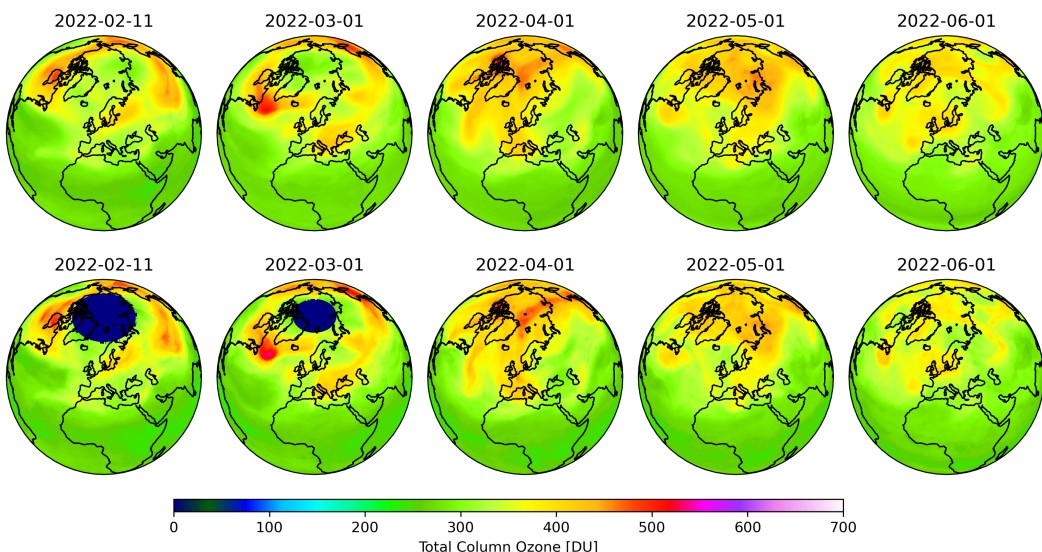

**Figure 3.** Comparison of Ozone Columns Calculated by LINOZ (top) and Satellite Data (bottom). The top row shows a model run that was initialized daily (start date 11th of February 2022, 40 km horizontal resolution, 90 height levels) with meteorological reanalysis data by DWD, while ozone was initialized from CAMS EAC4 data (Inness et al., 2019) on February 11, 2022 and then passed on. The bottom row shows satellite data provided by NASA Ozone Watch (NASA, n.d.).

- **Linoz v3**: To account for the impact of NOy on stratospheric ozone, Linoz version 3 (Hsu and Prather, 2010) was implemented into ART, using the linearized terms of temperature, $NO_y$, and ozone column.

- **Solar Spectral Irradiance** variability was incorporated by generating two distinct sets of LINOZ coefficients corresponding to solar maximum and solar minimum conditions. The model interpolates between these coefficient sets based on the daily F10.7 index (solar radio flux at 10.7 cm).

A comparison of NOy and ozone from model experiments with and without UBCNOy and with constant solar maximum respectively solar minimum SSI is provided in Figure 4, highlighting the impact of EEP-$NO_y$ on NOy, and the combined impact of EPP-$NO_y$ and variable SSI on ozone. A more detailed description and evaluation against observations are given in Ramezani Ziarani et al. (2025).

### 3.1.5 Stratospheric $N_2O$-$NO_y$ scheme

Tropospheric $N_2O$ is the main source of reactive nitrogen compounds in the stratosphere, which are summarised as $NO_y$ (Seinfeld and Pandis, 2016): $NO_y = NO + NO_2 + NO_3 + HNO_3 + HNO_4 + 2 N_2O_5 + ClONO_2 + ...$





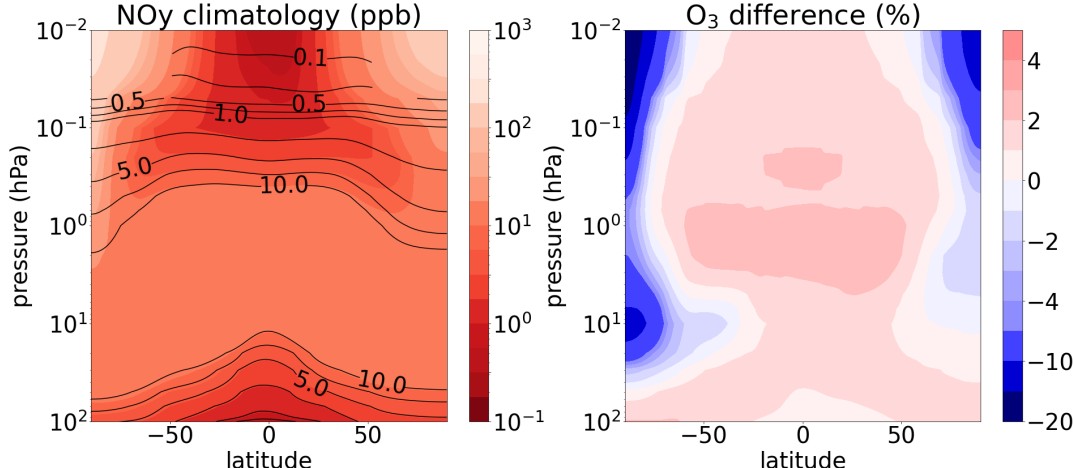

**Figure 4.** Left: NOy averaged over 8 years of model time. Colors: model experiment with variable EPP-NOy for 2002-2009 and constant solar maximum SSI. Lines: no EPP-NOy and constant solar minimum SSI. Right: percentage difference of ozone for the same period and model experiments, highlighting enhanced ozone formation around the tropical stratopause for solar maximum conditions, as well as strong ozone loss in polar regions due to the EPP-NOy. Model experiments following Ramezani Ziarani et al. (2025) using ICON-ART version 2025.04.

$N_2O$ is chemically inert in the troposphere and very long-lived due to its estimated lifetime of 120 years (Prather et al., 2015), so that it can be transported into the stratosphere. 90% of the stratospheric $N_2O$ is destroyed via photolysis:

$$N_2O + hv \rightarrow N_2 + O^1(D) \tag{R1}$$

The remaining $N_2O$ molecules react with excited oxygen atoms $O^1(D)$, which originate from ozone photolysis:

$$N_2O + O^1(D) \rightarrow N_2 + O_2 \tag{R2}$$

$$N_2O + O^1(D) \rightarrow NO + NO \tag{R3}$$

Reaction R3 leads to the production of two NO molecules, which initiate the $NO_y$ cycle through photolysis and oxidation:

$$NO + hv \rightarrow N + O^3(P) \tag{R4}$$

$$N + O_2 \rightarrow NO + O^3(P) \tag{R5}$$

$NO_y$ is destroyed via

$$N + NO \rightarrow N_2 + O^3(P) \tag{R6}$$





$$N + NO_2 \rightarrow N_2O + O^3(P) \tag{R7}$$

Since the production and destruction of NO according to the reactions R3 and R5 - R6 are in a first approximation the only chemical sources and sinks for $NO_y$, the simulation of $NO_y$ with $N_2O$ as a source takes place via the reactions R1 - R7 (Olsen et al., 2001).

The $N_2O$-$NO_y$ scheme of Olsen et al. (2001) is based on five parameters, which were determined in a photochemical box model at 20 pressure altitudes between 14 - 52 km, 18 latitudes between $85°S$ - $85°N$ and for 12 months and are available as parameter tables:

- C1: 24h - mean of $N_2O$ loss frequency

- C2: 24h - mean of NO photolysis frequency per $NO_y$ molecule

- C3: proportion of NO formation during $N_2O$ degradation

- C4: proportion of N reacting with NO or $NO_2$

- C5: $N_2O$ production rate

In order to interpolate the coefficients onto the ICON grid, the nearest neighbour method is used for horizontal interpolation and linear interpolation is used for vertical interpolation based on geometric height. As the coefficients are only defined between the heights of 14 - 52 km, the corresponding boundary values at 14 and 52 km are used for the layers below and above.

## 3.2 Detailed chemistry mechanisms

ICON-ART supports comprehensive and scalable gas-phase chemistry simulations through the integration of the atmospheric chemistry module MECCA (Module Efficiently Calculating the Chemistry of the Atmosphere), which is part of the CAABA (Chemistry As A Boxmodel Application) framework (Sander et al., 2019). MECCA provides a flexible and extensible platform for representing detailed chemical processes in the troposphere and stratosphere, including oxidation pathways, radical chemistry, and heterogeneous reactions.

The numerical integration of the chemical system is performed using the Kinetic Pre-Processor (KPP) (Sandu and Sander, 2006), which generates optimized code for solving systems of ordinary differential equations representing chemical kinetics. ICON-ART uses the Rosenbrock solver (Sandu et al., 1997) within KPP for efficient and stable time integration, particularly suitable for stiff chemical systems.

Photolysis rates, a critical component of atmospheric chemistry, are computed using CloudJ (Prather, 2015), an advanced photolysis scheme that accounts for the effects of clouds, aerosols, and molecular absorption in a multi-wavelength approach. CloudJ can be used in offline or online configurations, depending on the application and computational cost considerations.




A key feature of the ICON-ART tracer framework is the use of MECCA as an external preprocessor, which enables users to define and compile custom chemical mechanisms tailored to specific scientific questions or case studies. This modular approach allows for the integration of predefined standard mechanisms as well as user-defined chemical schemes. The integration of MECCA into ICON-ART has been previously described in detail by Schröter et al. (2018). In this section, we summarize the available chemistry mechanisms currently implemented in ICON-ART and provide guidance on their configuration and
application domains.

### 3.2.1 MOZART-4 Chemistry

In CAABA/MECCA version 4.0 (available in the supplementary material of Sander et al. 2019), the MOZART-4 chemical mechanism (Model for Ozone and Related chemical Tracers, version 4) from Emmons et al. (2010) has been integrated. Figure 5 presents a schematic overview of the steps required to run an ICON-ART simulation with MOZART-4 chemistry.
This includes processing the chemical reaction mechanism in MECCA and generating Fortran90 code via KPP for numerical integration. The implementation of MOZART-4 into ART utilizes additional scripts provided in the supplement. A detailed user guide for these preprocessing steps is available in Appendix C.

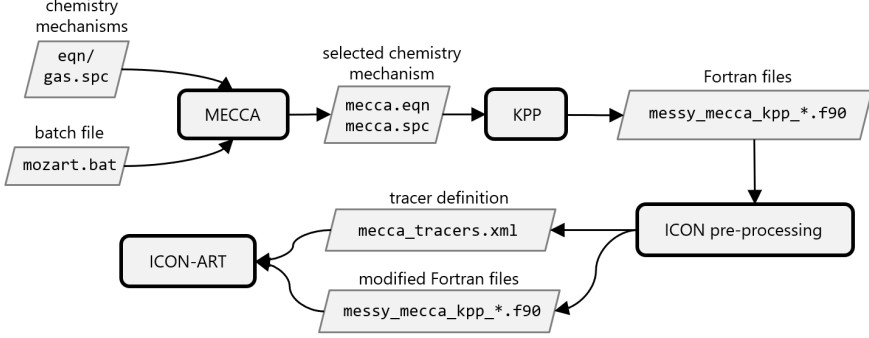

**Figure 5.** Schematic of the processing steps for integrating a custom chemistry mechanism into ART.

### 3.2.2 MOZART-T1 Chemistry

In Emmons et al. (2020), a new MOZART tropospheric chemistry scheme (MOZART-T1) was introduced, incorporating
several improvements over the previous version (MOZART-4), including improved oxidation of isoprene and terpenes, refined organic nitrate speciation, and more accurate aromatic speciation and oxidation, among others. As a result, MOZART-T1 provides an improved representation of ozone in the troposphere. For more details on how to implement the MOZART-T1 chemistry scheme in ART, the reader is referred to Appendix C.

     Figure 6 presents ground-level ozone pollution from a full chemistry simulation using the MOZART-T1 mechanism. The
figure shows the modeled mean afternoon ozone mixing ratios for winter and summer 2019, alongside observed values from the





EMEP monitoring network (European Monitoring and Evaluation Programme, data retrieved from NILU 2025). The ICON-ART simulations were conducted in limited-area mode over Europe using a R3B07 grid ($\Delta x \approx 13.2$ km). Initial and boundary conditions for trace gases and aerosols were derived from CAM-Chem output (Buchholz et al., 2024). Figure 6 demonstrates that the model accurately reproduces seasonal variations in ozone levels and effectively captures spatial variability.

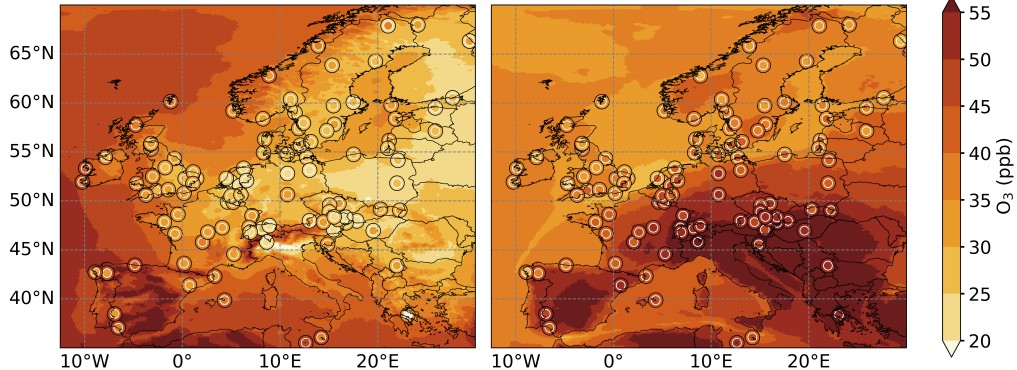

**Figure 6.** Mean afternoon ground-level $O_3$ mixing ratios from ICON-ART simulations for winter (JF, left) and summer (JJA, right) 2019. The inner filled dots represent observed values from EMEP measuring stations, while the outer rings indicate the corresponding simulated values co-sampled with the measurements. Mountain stations are excluded from this comparison.

### 3.3 Scheme for polar stratospheric clouds

Polar stratospheric clouds (PSCs) are an essential part for polar ozone depletion (e.g., Solomon, 1999), and therefore have to be accounted for in atmospheric chemistry models. In ICON-ART, the three major known types of PSCs are implemented. Nitric acid trihydrate (NAT) particles are calculated based on the scheme introduced by van den Broek et al. (2004) using a fixed measured size distribution where each size bin is transported as a passive tracer. Size changes are based on the saturation conditions for NAT particles, based on Hanson and Mauersberger (1988). The calculation of supercooled ternary solution (STS) particles uses the diagnostic scheme by Carslaw et al. (1995) with an adapted volume concentration (Hervig and Deshler, 1998). For ice particles, it has been found that lifting the altitude where the ICON microphysics are calculated to the lower stratosphere leads to realistic formation and transport of these particles (Weimer, 2019). Further details can be found in Weimer et al. (2021) where the scheme has also been applied to local grid refinements around the Antarctica.

### 3.4 Removal processes

#### 3.4.1 Parameterization of $CO_2$ deposition in the ocean

$CO_2$ is the major driver of anthropogenic climate change (Solomon et al., 2009), so that it should be accounted for in atmospheric chemistry simulation. One large sink of $CO_2$ is the deposition in the ocean which is implemented in ICON-ART in a simplified way. Jacob (1999) conclude that $50\%$ of the emitted $CO_2$ remains in the atmosphere, which we incorporated in



ICON-ART as a possibility for deposition in the ocean. We assume an average emission of $E_{CO2} = 2.77 \times 10^{-9} \mathrm{kg\,m^{-2}\,s^{-1}}$, divide this value by the fraction of ocean surface on Earth $r_{ocean} = 0.707$ and multiply it by the deposition factor $r_{depo} = 0.5$ based on Jacob (1999) to get the effective deposition $f_{eff,CO2}$ of $CO_2$ in the ocean:

$$f_{eff,CO2} = \frac{E_{CO2}\, r_{depo}}{r_{ocean}} \tag{6}$$

This deposition is then converted to mixing ratio and subtracted from the atmospheric $CO_2$ mixing ratio in the lowest model
layer.

The constant value of $E_{CO2}$ is based on the external emission datasets of the year 2012. It is increasing and should be adapted in the future to be time-dependent based on the current emissions of $CO_2$. However, taking global averages needed for this case will decrease the model performance, because all parallel processes have to wait for each other, which is why a constant value is used as a first approximation. Another possible improvement of this could be to use the yearly global growth
rates of $CO_2$. This deposition is automatically applied when the tracer is called TRCO2. Combined effect of this deposition and the non-globally-uniform lifetime is shown in Fig. 7.

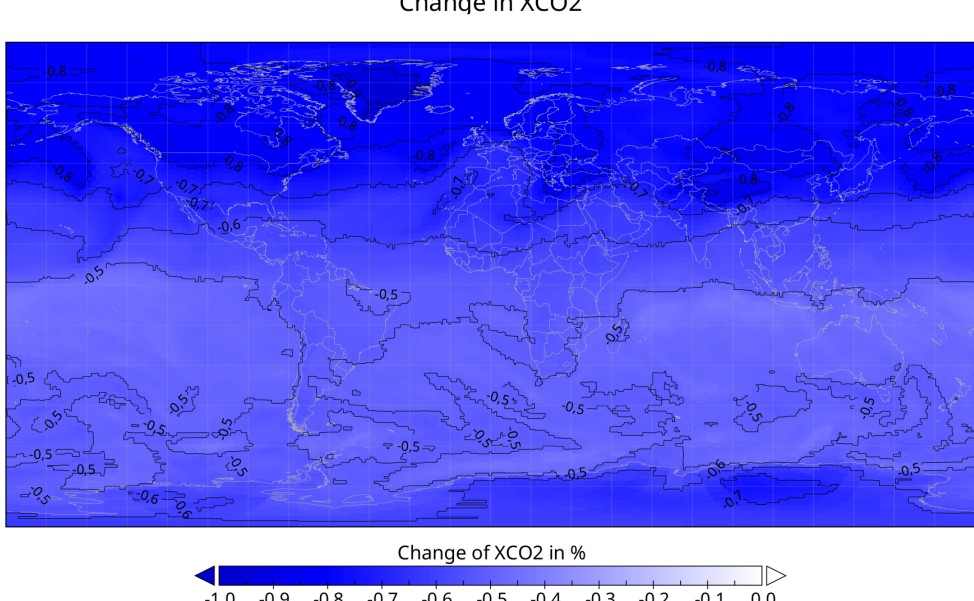

**Figure 7.** Percentage change in lifetime tracer for column-averaged carbon dioxide when tracer is called TRCO2 instead of CO2. Effects due to lifetime parameterization and deposition in the ocean after 2 years of simulation.





### 3.4.2 Dry deposition

The gaseous dry deposition parameterization follows the widely used resistance-in-series approach introduced by Wesely and Hicks (1977). This method expresses the deposition velocity as $v_d = -F/C(z_{\mathrm{ref}})$, where $F$ is the flux to the surface, and $C(z_{\mathrm{ref}})$ is the concentration at a reference height $z_{\mathrm{ref}}$. The deposition velocity is then modeled using resistances

$$v_d = \frac{1}{(r_a + r_b + r_c)} \quad , \tag{7}$$

where $r_a$, $r_b$ and $r_c$ are the aerodynamic, boundary layer and canopy resistances, respectively. The aerodynamic resistance $(r_a)$ is the same for all gases and depends on surface properties, wind speed, and atmospheric stability. The boundary layer resistance $(r_b)$ accounts for gas transport through the quasi-laminar layer to the surface and is influenced by the molecular diffusion coefficient $(D_i)$ of the chemical species $i$ in air (e.g. Seinfeld and Pandis, 2016).

The canopy resistance parameterization in ICON-ART follows Baer (1992), incorporating the influence of plant species composition, vegetation physiological state, and the chemical properties of the trace gases. Based on Baer (1992), ICON-ART distinguishes $n_{\mathrm{LU}} = 7$ deposition land use classes, each with tabulated plant specific constants and leaf area indices (LAI). The total canopy resistance is calculated as a combination of stomatal $(r_{\mathrm{st}})$, mesophyll $(r_{\mathrm{mes}})$, cuticular $(r_{\mathrm{cut}})$, and soil $(r_{\mathrm{soil}})$ resistances, as described in Equation (10) below.

The uptake of trace gases through stomata occurs by diffusion and is therefore inversely proportional to their molecular diffusivities. In ICON-ART, stomatal opening is influenced by photosynthetically active radiation $(I_{\mathrm{PAR}})$ and temperature, following the multiplicative model by Jarvis (1976):

$$r_{\mathrm{st}} = r_{\mathrm{st,min}} \left( 1 + \frac{b}{I_{\mathrm{PAR}}} \right) \frac{1}{f_T f_D} \quad . \tag{8}$$

Here, $r_{\mathrm{st,min}}$ is an empirically determined minimum resistance (Körner et al., 1979), and $b$ an empirical constant, both tabulated for each deposition land use class. The correction factor $f_T$ accounts for temperature effects, particularly stomatal closure at excessively low or high temperatures. The term

$$f_D = \frac{D_i}{D_0} \tag{9}$$

represents the dependence on the diffusivity $D_i$ of the trace gas, where $D_0$ is the molecular diffusivity of the reference species for which $r_{\mathrm{st,min}}$ was determined (usually $H_2O$ or $CO_2$). Once inside the stomata, trace gases are either deposited onto the hydrated surface of mesophyll cells or undergo chemical destruction. These processes are represented by the mesophyll resistance $r_{\mathrm{mes}}$, which thus depends on the solubility and reactivity of the gas. Total stomatal resistance is then given by the sum $r_s = r_{\mathrm{st}} + r_{\mathrm{mes}}$. In ICON-ART, two additional uptake processes are considered: direct deposition onto the leaf cuticle and deposition to soils, represented by $r_{\mathrm{cut}}$ and $r_{\mathrm{soil}}$, respectively. Both processes depend on the solubility and reactivity of the trace gases and are parametrized using empirical values for $SO_2$ and $O_3$. The total canopy resistance in each grid cell is computed as

$$r_c = \sum_{j=1}^{n_{\mathrm{LU}}} \frac{x_j}{\left( \frac{1}{r_s(j)} + \frac{1}{r_{\mathrm{cut}}(j)} \right) \mathrm{LAI}(j) + \frac{1}{r_{\mathrm{soil}}(j)}} \quad , \tag{10}$$





where $x_j$ represents the land use fraction. This formula applies to any trace gas species, except for $H_2SO_4$, $SO_2$ and $O_3$, for which adapted parametrizations are implemented.

### 3.4.3 Wet deposition

The removal of trace gases from the atmosphere via wet deposition is typically divided into two processes: in-cloud and below-cloud scavenging. In ICON-ART, both processes are parameterized using scavenging ratios, following the formulation by Simpson et al. (2012). Specifically, the in-cloud scavenging of a soluble trace gas with mixing ratio $\chi$ is described by

$$\frac{d\chi}{dt} = -\chi \frac{W_{\text{in}} P}{\Delta z} \frac{1}{\rho_{\text{W}}}, \tag{11}$$

where $W_{\text{in}}$ is the in-cloud scavenging ratio, $P$ is the surface precipitation rate (kg m$^{-2}$ s$^{-1}$), $\Delta z$ is the scavenging depth (assumed to be 1000 m), and $\rho_{\text{W}}$ is the density of water. Below-cloud scavenging is treated analogously, with the in-cloud ratio $W_{\text{in}}$ replaced by the below-cloud scavenging ratio $W_{\text{out}}$. Scavenging ratios for the main soluble trace gases are provided in the supplementary material of Simpson et al. (2012).

## 4 Aerosol processes

Aerosol processes in ART are represented using a flexible log-normal modal framework (Rieger et al., 2015). Each mode can belong to the Aitken, accumulation, coarse, or giant size ranges, with mean diameters spanning from below 0.01 µm to above 10 µm. ART allows both externally (single-component) and internally mixed (multi-component) modes. The prognostic equations for number density ($\hat{\Psi}_{0,l}$) and mass mixing ratio ($\hat{\Psi}_{3,l}$) are solved at every fast physics time step:

$$\frac{\partial \left( \bar{\rho}_a \hat{\Psi}_{0,l} \right)}{\partial t} = -\nabla \cdot \left( \hat{v} \bar{\rho}_a \hat{\Psi}_{0,l} \right) - \nabla \cdot \left( \overline{\rho_a v'' \Psi''_{0,l}} \right) - \frac{\partial}{\partial z} \left( v_{\text{sed},0,l} \bar{\rho}_a \hat{\Phi}_{0,l} \right) - Wa_{0,l} - Ca_{0,l} - Nu_{0,l} - Em_{0,l} \tag{12}$$

$$\frac{\partial \left( \bar{\rho}_a \hat{\Psi}_{3,l} \right)}{\partial t} = -\nabla \cdot \left( \hat{v} \bar{\rho}_a \hat{\Psi}_{3,l} \right) - \nabla \cdot \left( \overline{\rho_a v'' \Psi''_{3,l}} \right) - \frac{\partial}{\partial z} \left( v_{\text{sed},0,l} \bar{\rho}_a \hat{\Phi}_{3,l} \right) - Wa_{3,l} - Ca_{3,l} - Nu_{3,l} - Em_{3,l} - Co_{3,l} - Ch_{3,l} - Eq_{3,l} \tag{13}$$

where $\rho_a$ is the air density, $\nabla \cdot \left( \hat{v} \bar{\rho}_a \hat{\Psi}_{M,l} \right)$ and $\nabla \cdot \left( \overline{\rho_a v'' \Psi''_{M,l}} \right)$ denote the changes of the $M$-th moment of mode $l$ due to advection and turbulent fluxes, respectively, $\frac{\partial}{\partial z} \left( v_{\text{sed},0,l} \bar{\rho}_a \hat{\Psi}_{M,l} \right)$ describes the sedimentation flux with $v_{\text{sed},M,l}$ being the sedimentation velocity, $Wa_{M,l}$ denotes washout due to wet scavenging below-cloud, $Nu_{M,l}$ denotes the nucleation, and $Em_{M,l}$ denotes the emission flux. The terms $Co_{3,l}$, $Ch_{3,l}$ and $Eq_{3,l}$ are relevant for the third moment only and denote condensation, chemical transformation and equilibrium gas-aerosol partitioning, respectively. The aerosol dynamics processes (nucleation, coagulation, condensation, chemical transformation and gas-aerosol partitioning) are relevant when considering internally mixed aerosols and are briefly explained in the following section. hats and overbars Dry deposition is considered as a lower





boundary condition for tracer transport in the lowest model layer. The surface flux is determined using a one-dimensional turbulence scheme, where dry deposition is represented through a parameterized deposition velocity (Rieger et al., 2015).

The standard deviation of the modes is kept constant during the whole simulation. The median diameter of modes can change during atmospheric transport and be diagnosed from the zeroth and third moment (Rieger et al., 2015; Muser et al., 2020).

## 4.1 Aerosol types and properties

It is crucial for an aerosol model to account for key dimensions of aerosol variability, such as chemical composition, size distribution, and mixing state (Riemer, 2002). ART is capable of representing these aspects, allowing for a comprehensive
description of aerosol interactions and impacts. It considers a diverse range of aerosol types, including carbonaceous particles, mineral dust, volcanic ash, sea salt, and water-soluble species such as sulfates, nitrates, and ammonium. Table 3 categorizes these aerosols based on their properties such as size modes, mixing states, and solubility. It distinguishes between externally and internally mixed aerosols, with some species exhibiting a mixed state depending on atmospheric conditions. The listed aerosols span a range of size modes from Aitken to giant, reflecting their diverse sources and transport characteristics. While
sulfate, nitrate, ammonium, and water are fully or partially soluble and internally mixed, other aerosols, such as carbonaceous particles, dust, volcanic ash, and sea salt, tend to be less soluble and can exist in both external and internal mixtures. If the mixed model includes only soluble species, the composition is assumed to be volume-averaged. However, if insoluble species are present in the mixed mode, the model assumes a core-shell structure. This assumption is critical for the aerosol optical properties, which are addressed in section 5.1.

**Table 3.** Aerosol components considered in ICON-ART, along with their standard mode sizes, mixing states, and solubility. Choices of components, their modes and mixing state are flexible and can be user-defined.

| Aerosol Component | Modes | Mixing State | Solubility |
|---|---|---|---|
| Carbonaceous | Accumulation | External/Internal | Insoluble/Mixed |
| Mineral Dust | Accumulation, Coarse, Giant[1] | External/Internal | Insoluble/Mixed |
| Volcanic Ash | Accumulation, Coarse, Giant | External/Internal | Insoluble/Mixed |
| Sea Salt (NaCl) | Accumulation, Coarse, Giant | External/Internal | Soluble/Mixed |
| Sulfate ($SO_4^{2-}$) | Aitken, Accumulation, Coarse | Internal | Soluble/Mixed |
| Nitrate ($NO_3^-$) | Aitken, Accumulation, Coarse | Internal | Soluble/Mixed |
| Ammonium ($NH_4^+$) | Aitken, Accumulation, Coarse | Internal | Soluble/Mixed |
| Water ($H_2O$) | Aitken, Accumulation, Coarse | Internal | Soluble/Mixed |

[1] Note that giant here does not refer to giant dust particles with diameters $> 62.5\,\mu m$ (Adebiyi et al., 2023).



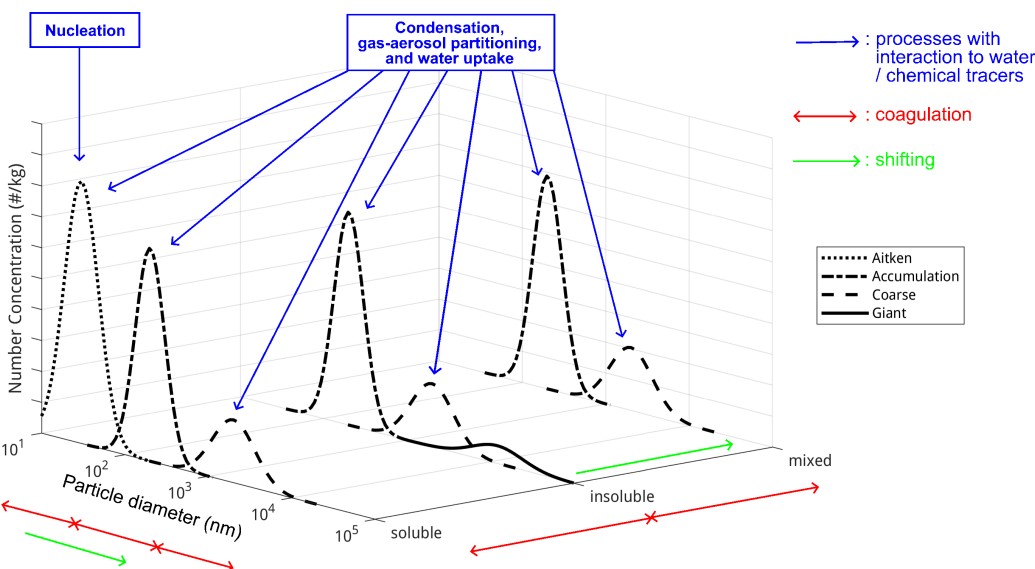

**Figure 8.** Schematic of the modes and aerosol dynamics processes in ICON-ART. Modes and processes can be configured by the user as needed, allowing for flexible configuration based on modeling requirements.

### 4.2 Aerosol dynamics

The ICON-ART model considers aerosol dynamical processes using its AERODYN module (Muser et al., 2020), which simulates key processes such as nucleation, condensation, coagulation, and gas-aerosol partitioning. AERODYN allows to consider a flexible number of log-normal modes (up to 10) to represent various aerosol sizes (Aitken, accumulation, coarse) and three different mixing states (soluble, insoluble, mixed) as well as an insoluble giant mode, which is not involved in the aerosol dynamic processes. Table 3 summarizes all possible compounds in the different modes. As aerosol dynamics changes the sizes and mixing states of particles, two mechanisms shift particles to other modes: (1) a shift to larger modes, when a threshold diameter is exceeded, and (2) a shift to the corresponding mixed mode when the soluble components in the modes exceed a 5% mass threshold (Muser et al., 2020).

Nucleation forms new soluble Aitken particles from gaseous $H_2SO_4$ increasing the zeroth and third moment of the Aitken mode. The parametrization used in AERODYN is based on Kerminen and Wexler (1995). It calculates a critical $H_2SO_4$ concentration which depends on temperature and relative humidity and above which it is assumed that $H_2SO_4$ nucleates. In addition to forming new particles through nucleation, $H_2SO_4$ can condense in all modes (except the insoluble giant mode) leading to an increase of the third moment of a mode and a particle growth. The condensation is parameterized based on Whitby et al. (1991) and was adapted in ART from Riemer (2002).

Intermodal and intramodal coagulation can be activated between all aerosol modes. Coagulation increases the median diameter and reduces the total number concentration within the aerosol size distribution. The parameterizations for the coagulation rates of the zeroth and third moments are based on Riemer (2002) and references therein, particularly Whitby et al. (1991).





Although the coagulation matrix is flexible and can be user-defined, several key aspects should be considered regarding the assignment of particles to modes after coagulation:

– In the case of intramodal coagulation, particles remain within the same mode.

   – For intermodal coagulation, particles are assigned to the mode with the larger median diameter.

   – If a mixed mode is involved, the resulting particles are assigned to the mixed mode.

   – Coagulation between an insoluble and a soluble mode results in particles initially remaining in the insoluble mode. These particles may later be transferred to the mixed mode if the mass of soluble material exceeds a predefined threshold.

The equations for nucleation, condensation, and coagulation are given in Appendix A.

The ISORROPIA-2 model by Fountoukis and Nenes (2007) is coupled to ART to calculate gas-aerosol partitioning according to thermodynamics equilibrium. Its aerosol system considers the species potassium ($K^+$), sodium ($Na^+$), magnesium ($Mg^{2+}$), calcium ($Ca^{2+}$), ammonium ($NH_4^+$), sulfate ($SO_4^{2-}$), nitrate ($NO_3^-$), chloride ($Cl^-$), water ($H_2O$) and derives an equilibrium state for these species in the gas, liquid, and solid phase. The processes are called in the following order: coagulation, aerosol-
gas partitioning, condensation, nucleation, and shifting.

## 4.3  Aerosol water uptake

The water uptake by aerosols in the atmosphere depends on the ambient relative humidity and the chemical composition of the aerosol. The chemical composition is determined by the molality, which is the amount of a component per kg of solvent. As molality or relative humidity increases, the liquid water content also increases. Conversely, crystallization can occur when the
relative humidity falls below the efflorescence relative humidity (ERH). The relative humidity at which no further water uptake occurs depends on the aerosol components and their efflorescence properties. Therefore, determining the crystallization point for an internally mixed aerosol is significantly more complex compared to an externally mixed aerosol.

The thermodynamic model ISORROPIA-2 (Fountoukis and Nenes, 2007) is used to calculate the water uptake by aerosols. However, this comes with computational overhead, which is undesirable, especially when focusing solely on water uptake on
aerosols like sea salt. Due to the high hygroscopicity of sea salt aerosol, which increases aerosol particle mass and consequently affects processes such as removal processes and optical properties, an alternative method for water uptake has been implemented in ICON-ART. This alternative method, previously applied in the COSMO-ART, is described in Lundgren et al. (2013).

## 4.4  Subpollen particles (SPPs)

As explained in Section 2.6, ART employs a parameterization for pollen emission (based on EMPOL-parameterization, Zink et al. 2013) and uses it to forecast alder, birch, grass and ragweed pollen at MeteoSwiss and DWD operationally. Due to their large size, pollen are not considered in the majority of processes in ICON-ART. Although they seem to be quite efficient in nucleating ice or as cloud condensation nuclei, they do not reach relevant altitudes in sufficiently large numbers to actually impact



microphysical processes. In recent years, so-called subpollen particles (SPPs), released by those large pollen grains rupturing and bursting, have received increasing attention, since their significantly smaller size enables them to reach those altitudes in larger numbers. To reflect this increased interest, Werchner et al. (2022) implemented a pollen bursting parameterization into ICON-ART based on physical assumptions, observations and processes (according to Zhou (2014)), enabling it to emit SPPs.

The parameterization's driver is the turgor pressure that builds up inside each pollen grain. Once this turgor pressure reaches a pollen specific critical value that the pollen walls can no longer withstand the pollen bursts and releases SPPs into the atmosphere. The turgor pressure's temporal development is formulated to be:

$$p_{\mathrm{T}}(t_{\mathrm{m}}) = p_{\mathrm{a}} + \Delta\pi \left[ 1 - \exp\left( -\frac{3Ek}{r} \frac{\rho_{\mathrm{w}}}{\rho_0} t_{\mathrm{m}} \right) \right]. \tag{14}$$

In Eq. (14), $p_{\mathrm{T}}$ and $p_{\mathrm{a}}$ are the turgor pressure and the ambient pressure, respectively, in $\mathrm{Pa}$, $t_{\mathrm{m}}$ is the model time step in $\mathrm{s}$, $\Delta\pi$ is the difference in water potential in $\mathrm{Pa}$, $E$ is the compression module of water in $\mathrm{N\,m^{-2}}$, $k$ is the pollen grain's water permeability in $\mathrm{m\,s^{-1}\,Pa^{-1}}$, $r$ is the pollen grain's radius in $\mathrm{m}$ and $\rho_{\mathrm{w}}$ and $\rho_0$ are water and pollen density, respectively, in $\mathrm{kg\,m^{-3}}$.

A complete description of this parameterization is given in Werchner et al. (2022).

## 5 Interactions and feedback mechanisms

### 5.1 Aerosol-radiation interaction

ICON uses ecRad (Hogan and Bozzo, 2018) as the standard radiation scheme for numerical weather prediction (Rieger et al., 2019). To account for aerosol–radiation interactions (ARI), ART computes the local radiative transfer parameters based on the aerosol optical properties—mass extinction coefficient, single scattering albedo, and asymmetry parameter—as well as the prognostic aerosol mass concentrations at each grid point and model level. These parameters are then passed as input to the radiation scheme (Rieger et al., 2017).

In addition to the prognostic ARI calculations, ART includes forward operators to diagnose aerosol optical depth (AOD) and attenuated backscatter at different wavelengths. These diagnostics are derived by multiplying the prognosed aerosol mass concentrations with mass extinction and backscatter coefficients (Hoshyaripour et al., 2019).

ART provides two approaches for specifying aerosol optical properties. In the first, these properties are precomputed offline and stored in lookup tables for use in both prognostic and diagnostic calculations (Hoshyaripour et al., 2019; Muser et al., 2020). This method accounts for the variability in aerosol size distribution by applying polynomial fits to the optical properties (Gasch et al., 2017). However, it is limited in capturing variations in particle composition and mixing state (e.g., coating), particularly when aerosol dynamics lead to the formation of internally mixed particles during simulations.

To address these limitations, the second approach computes aerosol optical properties online using MieAI (Kumar et al., 2024), enabling a more flexible and physically consistent representation. Further details on both approaches are provided below.





### 5.1.1 Prescribed aerosol optics

In ICON-ART, the prescribed optical properties for aerosols were originally implemented as hard-coded tables within the model code (Gasch et al., 2017; Rieger et al., 2017). This static approach limited flexibility and made updates or extensions to the optical property datasets cumbersome. To overcome these limitations, the optical properties have been transitioned to external NetCDF files, allowing for a more modular and user-configurable setup. This change enables users to easily update or switch between different aerosol optical property datasets without modifying the source code, facilitates consistency across simulations, and supports the use of more detailed and physically representative optical data, including properties for varying size distributions, compositions, and wavelengths.

The NetCDF database consists of extinction coefficients, single scattering albedo values, asymmetry parameters, and backscattering coefficients in three different wavelength groups:

- 30 prognostic wavebands, as used by ecRad

- 9 diagnostic wavelengths from AERONET: 340.0, 380.0, 440.0, 500.0, 550.0 , 675.0, 870.0, 1020.0 and 1064.0 nm

- 3 diagnostic wavelengths for lidar (ceilometer or satellites): 355.0, 532.0 and 1064.0 nm

An overview of all available modes and their microphysical properties can be found in Appendix B.

### 5.1.2 Online aerosol optics with MieAI

Accurate and efficient estimation of aerosol-radiation interaction is paramount for precise weather and climate prediction. The interaction between aerosols and radiation is governed by their optical properties, which are determined by various aerosol attributes such as morphology, size distribution, and chemical composition. These properties undergo significant changes due to chemical reactions and microphysical processes as aerosol particles are transported through the atmosphere. Traditional methods for computing aerosol optical properties rely on precomputed look-up tables (LUTs), which are computationally inexpensive but prone to substantial errors.

To address this limitation, we have integrated MieAI, a recently developed neural network-based approach, to compute the optical properties of internally mixed aerosol particles with ICON-ART (Kumar et al., 2024). MieAI is a fully connected feed forward neural network with 4 hidden layers, each layer having 64 neurons and uses Gaussian Error Linear Unit (GELU) as the activation function. It emulates Mie theory, enabling on-the-fly optical properties calculations that account for variations in aerosol size distribution and chemical composition. MieAI takes aerosol mass concentration of aerosol components from ICON-ART simulations as input and outputs key optical properties, including the mass extinction coefficient, single scattering albedo, and asymmetry parameter. These estimated optical properties are then used in radiative transfer calculations to calculate the radiative effects of aerosols more efficiently and accurately. Fortran-Keras bridge (FKB) was used for online coupling of MieAI with ICON-ART (Ott et al., 2020). See Kumar et al. (2024) for detailed discussion about MieAI.

An example usecase is shown in Fig. 9 for an ICON-ART simulation involving the La Soufrière volcanic eruption event in April 2021. Here, Fig. 9a shows the net shortwave flux at the surface estimated using traditional LUT approach whereas



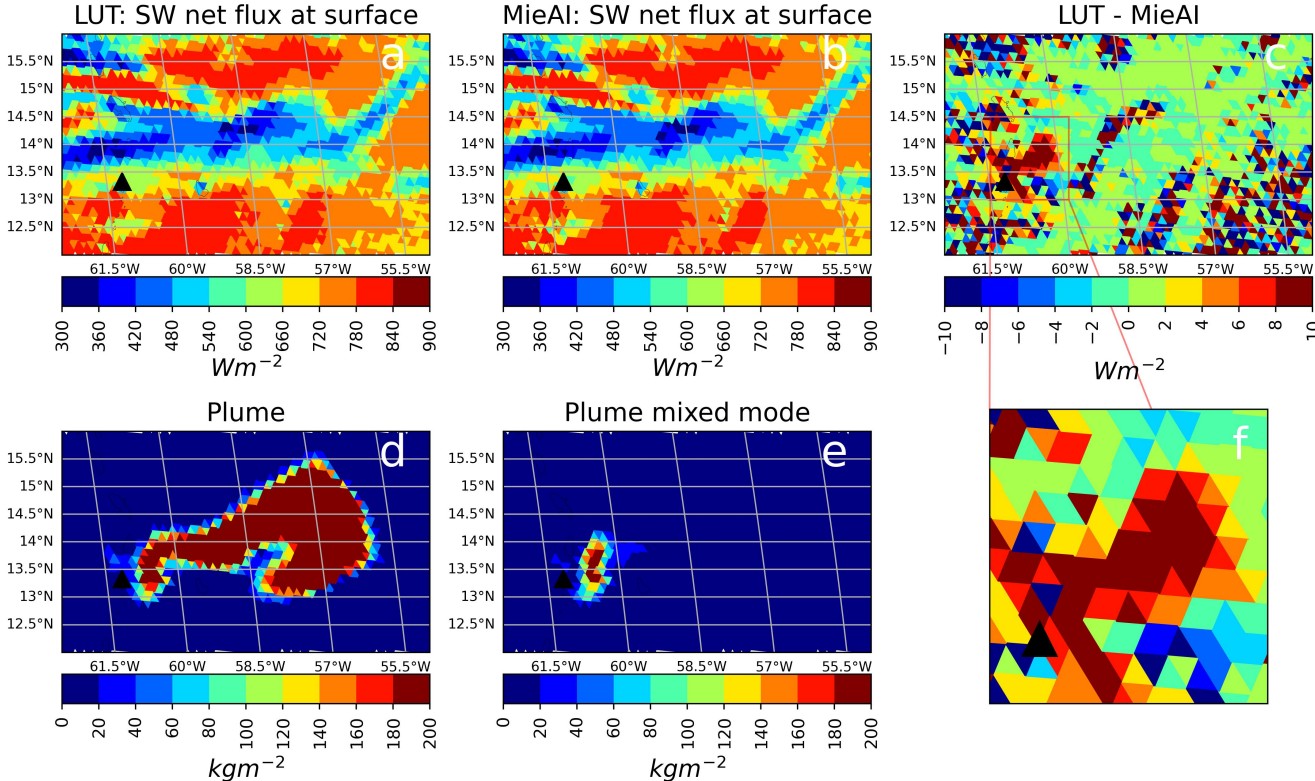

**Figure 9.** Comparison of net shortwave radiative flux estimated using MieAI against those estimated using Look-up table (LUT) approach for a case study involving the La Soufrière volcanic eruption (denoted by the black triangle) event simulated using ICON-ART. Here, panel **a)** shows the net SW flux estimated using LUT, **b)** shows the same estimated using MieAI and **c)** shows the absolute difference between them. The volcanic plume is depicted in panel **d)** whereas panel **e)** shows the mixed mode aerosols within the plume. Panel **f)** zooms panel **c)** over the plume region.

Fig. 9b shows the same using MieAI apporach and the difference between them are shown in Fig. 9c. As can be seen clearly from Figs. 9c and 9f, MieAI estimates lower SW flux ($> 8\,\mathrm{Wm}^{-2}$) at the surface over the region containing the volcanic plume depicted in Fig. 9d. A large part of this reduction in SW flux is due to the presence of mixed modes in the plume captured realistically by MieAI, causing enhanced extinction of solar radiation reaching the surface.

### 5.1.3 Multiple radiation call

Apart from including aerosol-radiation interactions in the modeling of atmospheric dynamics and composition, its diagnostic quantification is key to understanding aerosol impacts. To quantify the direct radiative effect (DRE), it is important to impose a technique that does not alter meteorological conditions, which would induce feedback processes and ultimately modify the aerosol effect. To diagnose the DRE, we therefore implemented a radiation multiple call scheme in ICON-ART, as it is also





implemented in other models (Woodward, 2001; Balkanski et al., 2007; Zhao et al., 2013; Klose et al., 2021). This approach executes calls of the radiation routine not just once (as needed to include aerosol-radiation interactions in the simulation), but multiple times for diagnostic purposes. In the first call, all aerosols are neglected and the radiation fluxes are stored. In optional intermediate calls, the effect of a single aerosol mode (and thus aerosol species) on the DRE can be quantified by

either including or omitting the respective mode (see below). In the last call, all aerosol-radiation interactions are accounted for, just as in the conventional model run. Therefore, the model integration is not impacted by execution of the multiple call scheme. The aerosol DRE is then directly obtained as the difference of the radiation fluxes between the two (or multiple) calls.

In ICON-ART, three different types of the radiation multiple call are available, following the concept implemented in Klose et al. (2021):

**Type I – Double call:** two calls, one without aerosols, one with all aerosols

**Type II – Inclusive multiple call:** first call without aerosols, one call per mode considering only that mode, last call with all aerosols

**Type III – Exclusive multiple call:** first call without aerosols, one call per mode considering all aerosols *except* that mode, last call with all aerosols.

The double call includes in total $n = 2$ calls per radiation timestep. The inclusive and exclusive multiple calls entail $n = n_{\mathrm{modes}} + 2$ calls, with $n_{\mathrm{modes}}$ being the number of aerosol modes, as for each mode there is one intermediate call. The DRE for each mode depends on the aerosol distribution of all modes in the column, such that the total DRE for all modes is not exactly equal to the sum of the individual contributions per mode, especially in regions with high aerosol load. Thus, call types II and III yield similar, but not equal results, especially in those regions. As call type III includes effects of other aerosol modes, we

consider this type preferential over the conceptually simpler type II.

For the first two call types (double call and inclusive multiple call), the DRE for a specific mode m is evaluated for both short-wave (wb=sw) and longwave (wb=lw) radiation, and at the surface (lev=sfc) and the top of atmosphere (lev=toa), $DRE_{\mathrm{wb}}^{\mathrm{m}}|_{\mathrm{lev}}$. It is calculated from the radiative flux $F_{\mathrm{wb}}^{\mathrm{m}}|_{\mathrm{lev}}$ as

$$DRE_{\mathrm{wb}}^{\mathrm{m}}|_{\mathrm{lev}} = F_{\mathrm{wb}}^{\mathrm{m}}|_{\mathrm{lev}} - F_{\mathrm{wb}}^{\mathrm{none}}|_{\mathrm{lev}} \,, \tag{15}$$

where $F_{\mathrm{wb}}^{\mathrm{none}}|_{\mathrm{lev}}$ denotes the flux without any aerosol-radiation interactions, obtained from the first call. For call type III, the exclusive multiple call, the DRE for all aerosols is calculated in the same manner as

$$DRE_{\mathrm{wb}}^{\mathrm{all}}|_{\mathrm{lev}} = F_{\mathrm{wb}}^{\mathrm{all}}|_{\mathrm{lev}} - F_{\mathrm{wb}}^{\mathrm{none}}|_{\mathrm{lev}} \,. \tag{16}$$

To obtain the DRE for individual modes, the situation is slightly different for type III. The flux obtained from the call for a specific mode m, $F_{\mathrm{wb}}^{\mathrm{all\text{-}m}}|_{\mathrm{lev}}$, includes the effect of all modes except the desired mode. Thus, the reference flux for the DRE in

this case is the one including the effect of all aerosols, yielding

$$DRE_{\mathrm{wb}}^{\mathrm{all}}|_{\mathrm{lev}} = F_{\mathrm{wb}}^{\mathrm{all}}|_{\mathrm{lev}} - F_{\mathrm{wb}}^{\mathrm{all\text{-}m}}|_{\mathrm{lev}} \,. \tag{17}$$





All calculated DREs are available as output, yielding four diagnostic variables (2 wavebands and two levels) for the double call (type I) and another four per mode for the multiple calls, e.g. 16 for the case of a multiple call with the three default dust modes $dust_{acc}$, $dust_{coa}$ and $dust_{gia}$ (cf. Tab. 3). They are available both as instantaneous values and as accumulated quantities. For the latter, note that the accumulation occurs between times at which the DRE is evaluated, i.e. with the radiation time step.

An example of the short wave DRE for dust is presented in Fig. 10. Shown is the annual average DRE for all three default dust modes (Fig. 10a-c), and the total DRE for all modes (Fig. 10d), obtained using the exclusive multiple call. As expected, the main contribution to the short wave dust DRE stems from the accumulation dust mode, which yields a strong cooling effect of up to $8.4\,\mathrm{W\,m^{-2}}$ near the horn of Africa. In contrast, the giant dust mode features a positive short wave DRE at the top of the atmosphere over land in areas of high dust loading due to the stronger absorption. Dust in the coarse mode contributes with an intermediate signal to the total DRE, which is overall negative in our example. Note that the size mode terminology used here follows the mode definitions given in Tab. 3 and differs from the dust size classification proposed in Adebiyi et al. (2023).

## 5.2 Aerosol-cloud interaction

To investigate aerosol–cloud interactions (ACI), ART is coupled with the two-moment cloud microphysics scheme of ICON (Seifert and Beheng, 2006), which provides a detailed and physically consistent representation of cloud processes. This scheme includes six hydrometeor categories—cloud droplets, ice crystals, rain, snow, graupel, and hail—and predicts both their mass and number concentrations through a set of prognostic budget equations. The evolution of liquid-phase hydrometeors is governed by processes such as condensational growth, autoconversion, accretion, self-collection, breakup, and freezing. In contrast, ice-phase hydrometeors are subject to diffusional growth, aggregation, self-collection, riming, secondary ice production (ice multiplication), and melting. The two-moment formulation enhances the model's ability to capture the sensitivity of cloud development and precipitation formation to aerosol perturbations, making it well-suited for studies of ACI (Rieger et al., 2017; Gruber et al., 2019).

### 5.2.1 CCN and IN activation

A physically-based parameterization for cloud condensation nuclei (CCN) activation, following Abdul-Razzak et al. (1998) and Abdul-Razzak and Ghan (2000), is implemented within the two-moment cloud microphysics scheme of ICON-ART. This approach enables a more realistic coupling between aerosols and cloud droplet formation by using grid-scale vertical velocity and a parameterized subgrid contribution, available moisture, and soluble aerosol concentrations to compute the number concentration and mass mixing ratio of newly formed cloud droplets. The activation is computed only under appropriate atmospheric conditions, such as the presence of updrafts and supersaturation with respect to liquid water. With this implementation, microphysical computations are removed from the ART module; ART now supplies the activation parameterization and outputs the newly formed cloud droplets, which are then further processed by ICON's native two-moment microphysics scheme Seifert and Beheng (2006). Initial testing and validation of the parameterization are performed successfully.

For heterogeneous ice nucleation, the calculation of the total surface area of all dust aerosol modes is fully implemented on the ART side and serves as the basis for determining the number of ice nucleating particles according to the ice nucleation active



**Figure 10.** Direct radiative effect (DRE) for shortwave radiation at the top of atmosphere. Panels show the DRE for the different dust modes, i.e. **a)** accumulation mode, **b)** coarse mode, **c)** giant mode (see Tab. 3), and **d)** all aerosols. Note that the color scales differ between panels.

site (INAS) density approach (Hoose and Möhler, 2012), following the empirical parameterizations for immersion freezing and deposition ice nucleation (e.g., Ullrich et al., 2017). This enables the prognostic treatment of heterogeneous ice nucleation in accordance with the evolving aerosol size distribution and composition within the model. Coupling of this INAS-based ice nucleation treatment with the two-moment microphysical scheme of ICON Seifert and Beheng (2006) is currently in progress. Once completed, this coupling will allow for a consistent representation of aerosol-cloud interactions, including the formation

of ice crystals from mineral dust under mixed-phase and cirrus conditions.





### 5.2.2 Dusty cirrus parametrization

In addition to the ACI on the grid-scale, ICON-ART includes a sub-grid parametrization of dusty cirrus. This special parametrization is beneficial or even necessary because dusty cirrus clouds can form due to a small-scale mixing instability, which is difficult to represent in model configurations that are not eddy resolving (Seifert et al., 2023).

The dusty cirrus parametrization uses the mass concentration of mineral dust $c_{\mathrm{mode}}$ assuming three dust modes with mode $\in$ {dustA, dustB, dustC}. Humidity is quantified by the ice saturation ratio $s_{\mathrm{ice}} = p_v/p_{\mathrm{sat,ice}}$, where $p_v$ is the vapor pressure and $p_{\mathrm{sat,ice}}$ is the saturation vapor pressure over ice. Atmospheric stability is characterized by the temperature lapse rate

$$\gamma_k = \left.\frac{\partial T}{\partial z}\right|_k \approx \frac{T_k - T_{k+1}}{\Delta z} \tag{18}$$

Note that ICON uses top-down indices, i.e., level $k+1$ is below level $k$. Dusty cirrus occurs in model level $k$ if the following
conditions are fulfilled:

$$T_k \quad < 240 \text{ K} \tag{19}$$

$$\hat{c}_{\mathrm{dust},k} = \max_{j=k+1}^{k+N+1} \left( c_{\mathrm{dustB},j} + 2\,c_{\mathrm{dustC},j} \right) > c_{\mathrm{dust}}^* \tag{20}$$

$$\hat{s}_{\mathrm{ice},k} = \max_{j=k}^{k+N} s_{\mathrm{ice},j} > s_{\mathrm{ice}}^* \tag{21}$$

$$\hat{\gamma}_k = \min_{j=k}^{k+1} \gamma_j < \gamma^* \tag{22}$$

with empirically determined thresholds $c_{\mathrm{dust}}^* = 50 \ \mu\text{g kg}^{-3}$, $s_{\mathrm{ice}}^* = 0.7$, and $\gamma^* = -6.5$ K km$^{-1}$ and with $N = 4$ corresponding to a vertical depth of approximately 1500 m. That the scheme uses information from 4 layers below to predict dusty cirrus layers corresponds to the convective overturning and the mixing instability. Note that $c_{\mathrm{dust}}^*$ does not include that smallest dust mode dustA, and the largest mode dustC has double the weight of dustB. The fact that the larger dust modes, dustB and dustC, are better predictors for the occurrence of a dusty cirrus than dustA is consistent with the increased ability of large mineral dust
particles to act as INPs, whereas smaller particles are less relevant for the formation of ice clouds by heterogeneous nucleation (DeMott et al., 2010, 2015). For the sub-grid dusty cirrus a cloud fraction of one is assumed and the sub-grid ice water content is set to 80 mg m$^{-3}$ with a linear tapering at the boundaries.

Figure 11 shows the results of a global ICON-ART simulation of a Saharan dust event over Europe leading to dusty cirrus cloud formation. The simulation uses the R03B06 icosahedral grid with a equivalent grid spacing of approximately 20 km.
The satellite observations show an extended cirrus deck over central Europe with outgoing longwave radiation (OLR) of 200 W m$^{-2}$ and lower. Without the dusty cirrus parameterization underestimates the extent of the dusty cirrus cloud. With the sub-grid parameterization the simulation is improved but OLR is underestimated at the boundary of the cirrus cloud deck. This suggests that the simplistic threshold based parametrization needs to be further improved in the future.



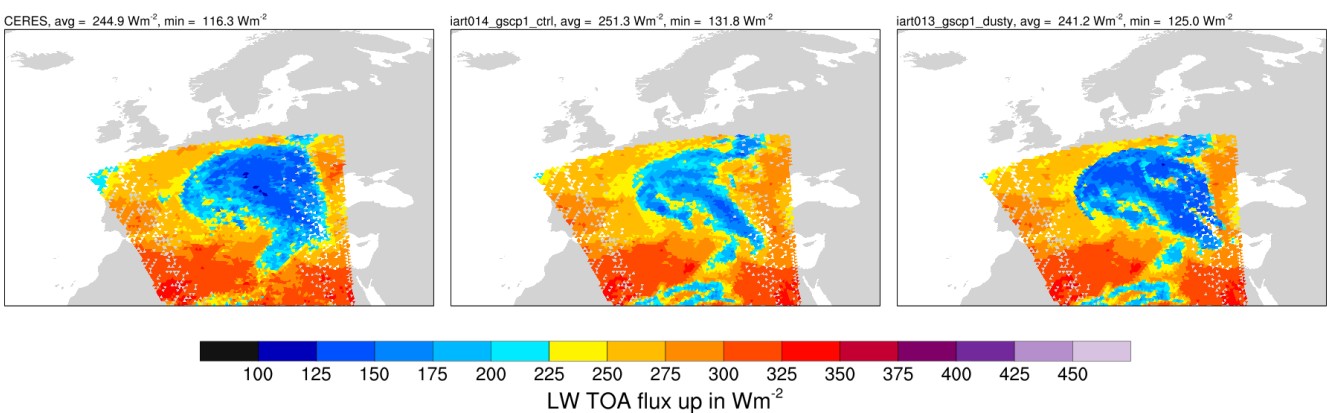

**Figure 11.** Comparison of global ICON-ART simulation for 12 UTC of 6 May 2022 with CERES Level 2 satellite data of outgoing longwave radiation at the top of atmosphere. Observations (left), ICON-ART without dusty cirrus parametrization (center), and ICON-ART with dusty cirrus parametrization (right).

## 6 Code infrastructure and periphery

### 6.1 Coupling ART with ICON

ART is currently designed as a submodel for ICON such that the development and organization of ART is independent from the base model in most aspects. Its coupling to ICON, however, is extensive, as Fig. 12 shows. Apart from several points within the time loop where ART provides additional processes, ART also influences some ICON processes directly. Beyond this process-based coupling, ART actively uses some structures introduced in the ICON code. The most prominent example for this is the tracer array, which ART extends to accommodate the additional tracers, whose set is defined by the user with ICON-ART's flexible tracer framework (Schröter et al., 2018). Accessing ICON's tracer array allows the ART tracers to be treated the same way as the base ICON tracers with regard to processes like advection or turbulent diffusion. Additionally, ART uses and extends structures that represent tracer metadata and requires grid information provided by ICON.

For the bulk of the communication between ICON and ART, a set of interface modules are used. The interfaces enable ICON to call process and other routines in ART altering the model state, and ART to retrieve crucial information and structures to fulfill its tasks. The separation of the different modules is based on the general stages of the model (examples: mo_art_init_interface.f90, mo_art_general_interface.f90 and mo_art_diagnostics_interface.f90) and the processes provided (examples: mo_art_sedimentation_interface.f90, mo_art_aerodyn_interface.f90, mo_art_coagulation_interface.f90 and mo_art_reaction_interface.f90). While previously the interfacing between ICON and ART was managed in the host model, it moved to the ART codebase in recent developments. This reduces the additional infrastructural burden when developing ART, since the ICON codebase is only altered if actually required.





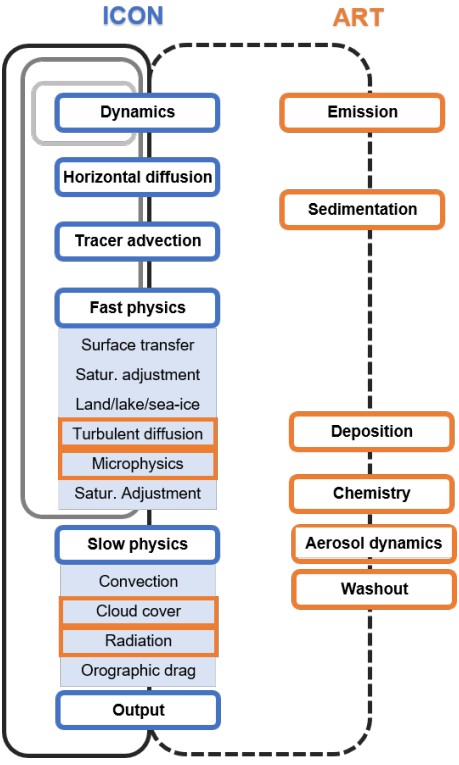

**Figure 12.** Schematic of the coupling of ICON–ART. The sequence in which processes of ICON are executed is illustrated by the blue boxes. Processes of ART are illustrated by the orange boxes. An orange frame around a blue box indicates, that the according code is part of the ICON tracer framework but ART tracers are treated inside this framework. The gray and black circles indicate the sequences of the time integration.

## 6.2 Flexible tracer and mapping framework

ART showed great success using its flexible tracer framework introduced by Schröter et al. (2018) in such a way that further development was conducted to extend this flexibility to processes. One of these processes is coagulation, which requires a
690 coagulation-matrix describing potential mode shifts when coagulation occurs. This matrix is to be provided to ART via a corresponding XML-file, while also adding a new XML-tag for the tracers indicating whether or not they should be considered for coagulation. An important part for the consideration of aerosols and tracers is their emission into the atmosphere. Emission parameterizations were initially assigned hardcoded by their and the corresponding tracers' name. This put a strain to the actual flexibility achieved, so a new method was introduced. By storing information (like number of emission fluxes or sizes of the
695 emitted material) about a specific emission parameterization and assigning substances to it in a new XML-structure, a mapping routine can map the emission fluxes to the correct mode of the specific substance before calling the emission parameterization itself. Also the new Online Emission Module uses XMLs to control its behaviour in ART.The usability of the underlying





XML-structure for providing the model with information also goes beyond the actual tracers, modes and processes, since also diagnostic variables can be defined with it.

### 6.3  GPU and exascale readiness

Since the development of ART has been largely driven by scientific motivations, particularly in the implementation of new processes and features, the resulting historically grown codebase contains numerous cross-dependencies between modules. This complexity makes it difficult to trace the code flow and hinders further development and maintenance. To address this issue, modularization efforts have been undertaken in recent years, streamlining the codebase, reducing inter-module dependencies, and providing a more intuitive interface. An additional benefit of these modularization efforts is the simplified parallelization and the potential to extend ART for use on new heterogeneous and future exascale HPC architectures.

Due to both its scientific design and historically evolved codebase, most processes, features, and modules have been developed exclusively for CPU usage. Although CPU parallelization is implemented and widely used across large parts of ICON, ART rarely exhibits comprehensive parallelization of its features. In recent years, the use of GPUs has gained increasing interest in scientific fields and is likely to continue doing so. Hence, more of the ART codebase needs to be ported to heterogeneous HPC architectures with GPUs to remain competitive and future-proof - due to its operational use at MeteoSwiss pollen-related ART-parts were already ported. As a first step, the online emissions module (OEM) and the vegetation photosynthesis and respiration model (VPRM) of ART have been refactored and ported using OpenACC compiler directives, following ICON's GPU porting approach in an attempt to accelerate the porting process while maintaining a unified codebase for both CPUs and GPUs. Inverse emission estimation with ICON-ART as first demonstrated by Steiner et al. (2024) and Thanwerdas et al. (2025) is now possible with a GPU-accelerated version of ICON-ART and has already been successfully tested and validated on Switzerland's new HPC system, ALPS, leveraging NVIDIA's Grace-Hopper superchips.

### 6.4  Standard configurations and simulation workflow

In ICON-ART, a set of standard experiment configurations has been established to ensure consistent testing and validation across model releases. Each configuration, see table 4, combines specific model components, tracer species, and physical or chemical processes relevant to targeted research or operational applications. For example, „Dust" focuses on mineral dust and its radiative feedback, while „Volcano" emphasizes volcanic ash and aerosol-radiation interactions. „Nat. Aero." incorporates a wide range of aerosol tracers and processes without radiative coupling, providing a comprehensive base for sensitivity studies. „Pollen" addresses both pollen and subpollen particles, representing bio-aerosols, and their relevant processes and the chemistry experiments („Simp. Chem." and „Comp. Chem.") are tailored for atmospheric composition forecasts, differing in the complexity of the chemistry scheme being used. „OEM" is configured for regional applications with full chemistry and online emission treatment. These standard configurations are fully supported, regularly maintained, and systematically tested with every ICON-ART release to ensure model integrity, reproducibility, and performance across use cases.

All standard experiments are also integrated into the workflow management solution *auto-icon* (Baer et al., 2025a, b), which enables users to orchestrate a simulation workflow covering the retrieval of input data, compilation, and execution of the model





as well as post-processing and visualization tasks. This integration enables a quick and smooth start for new users as well as easy-to-use and flexible testing and evaluation capabilities with the provided experiments during model development. In more complex model applications, *auto-icon* can significantly enhance throughput by executing all non-dependent tasks in parallel and wrapping individual tasks into larger allocations on the HPC system, which was shown to reduce the time-to-solution by

7 % or more (Marciani et al., 2025).

**Table 4.** Overview of standard ICON-ART configurations with the relevant tracers, and processes.

| Experiment | 1 | 2 | 3 | 4 | 5 | 6 | 7 |
|---|---|---|---|---|---|---|---|
| Note | Dust | Volcano | Nat. Aero. | Pollen | Simp. Chem. | Comp. Chem. | OEM |
| **Config.** | | | | | | | |
| Grid | 40 km | 40 km | 80 km | 6.5 km | 80 km | 16 km | 26 km |
| Nest | | ✓ | | | | | |
| LAM | | | | ✓ | | | ✓ |
| **Tracers** | | | | | | | |
| Mineral dust | ✓ | | ✓ | | | | |
| Bio-burning | | | ✓ | | | | |
| Sea salt | | | ✓ | | | | |
| Pollen | | | | ✓ | | | |
| Volcanic ash | | ✓ | | | | | |
| Radionuclide | | | | ✓ | | | |
| Region tracers | | | | | | | ✓ |
| Chem. tracers | | | | | | ✓ | |
| Mecca tracers | | | | | | | ✓ |
| **Processes** | | | | | | | |
| ARI | ✓ | ✓ | | | | | |
| Aerosol dynamics | | | ✓ | | | | |
| Chemistry | | | | | | ✓ | ✓ |
| **Anthropogenic emissions** | | | | | | | ✓ |

# 7    Conclusion and outlook

The current version, ICON-ART 2025.04, supports a wide range of applications, from air quality and weather forecasting to climate and Earth system simulations. It features a modular and flexible design, enabling the use of both simplified and detailed chemistry mechanisms, interactive aerosols, and online coupling with dynamics, radiation, and cloud microphysics.

Beside scientific research, ICON-ART is currently used in operational applications, including mineral dust forecasting at the Deutscher Wetterdienst (DWD), as well as pollen forecasting at both DWD and MeteoSwiss. These activities demonstrate the model's robustness, flexibility, and ability to support real-time decision-making in environmental and health-related services.

Ongoing developments focus on further enhancing the coupling between atmospheric composition and the physical core of ICON. This includes the integration of ART with the seamless ICON configuration (Müller et al., 2025) for unified weather



and climate simulations, which will allow for consistent aerosol and trace gas treatment across spatial and temporal scales. In addition, the implementation of organic aerosol processes is underway to improve the representation of both primary and secondary organic aerosol formation and their interactions with radiation and clouds.

Together, these efforts aim to strengthen ICON-ART as a versatile and comprehensive tool for studying the interactions between atmospheric composition and the Earth system in a fully coupled, scale-aware modeling framework.

*Code availability.* The ICON-ART model is open-source and accessible through the website: icon-model.org; ICON partnership (MPI-M; DWD; DKRZ; KIT; C2SM) (2024). ICON release 2025.04. WDCC at DKRZ. https://doi.org/10.35089/WDCC/IconRelease2025.04

The *auto-icon* project is open-source and accessible via Zenodo (Baer et al. (2025b)) and through the Gitlab repository: https://gitlab.com/auto-icon/auto-icon

*Data availability.* CERES SSF Level 2 data were obtained from NASA Langley Research Center (LaRC) Atmospheric Sciences Data Center
(ASDC) at https://asdc.larc.nasa.gov/data/CERES/SSF (link requires Earthdata login). After the completion of review processe, the output from ICON-ART simulations generated in this study will be uploaded on Radar4KIT. For any inquiries about the data from this study please contact Ali Hoshyaripour (ali.hoshyaripour@kit.edu).

*Author contributions.* The ICON-ART model is the result of a long-term collaborative effort. All authors contributed to the development, implementation, testing, or documentation of various components of the ICON-ART system. For this overview, G.A.H. and C.H. led the
writing of the manuscript, coordinated input from co-authors, and contributed to model development and evaluation. Section 1 (Introduction) was written by G.A.H. with major contributions from C.H. and P.L. Section 2 (Emissions) was led by D.B with contributions from M.S. (OEM), L.M. and N.P. (wildfires), G.A.H and J.F. (sea salt), J.B. (volcanic eruptions), M.W. (DMS), A.P. (pollen). Section 3 (Chemistry Processes) was lead by S.V. with the following contributors: L.R. (region tracers), V.H. and R.R. (lifetime and OH chemistry), M.S. and T. R. (Linoz v3 and UBCNOy), M.W. (Polar Stratospheric Clouds and $CO_2$ deposition) and C.K. (MOZART chemistry, dry deposition, wet
deposition). Section 4 (Aerosol Processes) was written by G.H.A with contributions from J.B (aerosol dynamics), H.V. (water uptake) and S.W. (SPP). Section 5 (Interactions and feedback mechanisms) was prepared by G.A.H and C.H. with contributions from E.M. (prescribed aerosol optics), P.K. (MieAI), A.B and M.K. (Multiple radiation call), S.B. (CCN activation), A.S. (Dusty cirrus parametrization). Section 6 (Code infrastructure and periphery) was lead by S.W. with contributions from A.H., A.B., M.K., K.S. and G.A.H. Section 7 (Conclusion and outlook) were written by G.A.H., C.H. and P.L. All authors reviewed the manuscript and provided substantial feedback.

*Competing interests.* The authors declare that they have no conflict of interest.



*Acknowledgements.* We thank the ICON development team at DWD, C2SM, DKRZ and MPI-M for their support of the ICON–ART development. This work contributes to and is partly funded by the following Deutsche Forschungsgemeinschaft (DFG) projects: Research Unit VolImpact (FOR2820, DFG Grant 398006378) subprojects VolPlume (HO 5275/4-2), OpenICON (HO 5275/6-1) and ZA 1208/1-1. This work used resources of the Deutsches Klimarechenzentrum (DKRZ) granted by its Scientific Steering Committee (WLA) under project ID 1070 and of the Swiss National Supercomputing Centre (CSCS) under project ID s1298. Part of this research has been funded by the project *PermaStrom* (grant no. 03EI4010A) within the seventh Energieforschungsprogramm of the German Federal Ministry of Economic Affairs and Climate Action (Bundesministerium für Wirtschaft und Klimaschutz, BMWK). Open Access funding enabled and organized by Projekt DEAL. The GPU porting of the ART components was supported by the Platform for Advanced Scientific Computing (PASC) project HAM and ART Acceleration for Many-Core Architectures (HAMAM). MK and AB acknowledge funding by the Helmholtz Association's Initiative and Networking Fund (grant agreement no. VH-NG-1533) and by the National High Performance Computing Center at Karlsruhe Institute of Technology (NHR@KIT), funded by the Ministry of Science, Research and the Arts Baden-Württemberg and by the Federal Ministry of Research, Technology and Space. CK acknowledges funding by the SCENE Joint Initiative co-financed by the ETH Board, Switzerland. This article has been supported by the use of ChatGPT, an AI language model developed by OpenAI, for grammar and phrasing corrections. The authors have critically reviewed and ensured the accuracy of all content. We thank Peter Braesicke and Bernhard Vogel for their valuable contributions to the development of ICON-ART during their time at KIT. Their leadership and foundational work have significantly shaped the model and its capabilities.



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



## Appendix A: Aerosol Dynamics

### A1  Nucleation

Based on the parametrization by Kerminen and Wexler (1995), new sulfate particles form when a critical concentration of $H_2SO_4$ (given in $\mu$g m$^{-3}$) is reached:

$$c_{crit} = 0.16 \exp\left(0.1T - 3.5\frac{RH}{100} - 27.7\right) \tag{A1}$$

with temperature T in K and relative humidity RH in %. The concentration above this threshold nucleates into the soluble Aitken mode with a mass rate of

$$Nu_{3,sol_A it} = \frac{c_{H2SO4} - \frac{c_{crit}}{\rho_p}}{\Delta T}. \tag{A2}$$

$c_{H2SO4}$ is the actual concentration of $H_2SO_4$. The number concentration (in kg$^{-1}$) follows from the assumed size distribution as

$$Nu_{0,sol_A it} = \frac{6}{\pi \rho_p} \cdot \frac{\exp\left(4.5 \cdot \ln^2(\sigma_{sol_A it})\right)}{d_{0,sol_A it}^3} \cdot Nu_{3,sol_A it}. \tag{A3}$$

### A2  Condensation

The condensation rate of the third moment of a mode $l$ is based on Whitby et al. (1991); Riemer (2002) and is given by

$$\tilde{C}o_{3,l} = \frac{6}{\pi}\chi_T \int\limits_0^\infty \chi(d_{3,0})\Psi_{3,l}(d)\mathrm{d}d_{3,l} = \frac{6}{\pi}\chi_T I_l. \tag{A4}$$

$\chi_T$ is independent of the particle size and depends on thermodynamic variables. It will be eliminated later. $\chi(d_{3,l})$ differs for different size regimes. The expression for the near-continuum and the free-molecular regime are given by

$$\chi_{nc}(d_l) = 2\pi D_{g,H2SO4}d_l \tag{A5}$$

and

$$\chi_{fm}(d_l) = \frac{\pi\alpha\bar{c}_{H2SO4}}{4}d_l^2 \tag{A6}$$

with $D_{g,H2SO4}$ the diffusion coefficient of $H_2SO_4$, $\alpha$ the accommodation coefficient, and $\bar{c}_{H2SO4}$ the mean molecular velocity of $H_2SO_4$. Using these two formulations, the integral $I_l$ in (A4) is evaluated separately as

$$I_l^{nc} = 2\pi D_{g,H2SO4}M_{1,l} \tag{A7}$$

and

$$I_l^{fm} = \frac{\pi\alpha\bar{c}_{H2SO4}}{4}M_{2,l}. \tag{A8}$$





With the harmonic mean of the expressions for the near-continuum and free-molecular regime, Eq. (A4) can be written as

$$\tilde{Co}_{3,l} = \frac{6}{\pi} \chi_T \frac{I_{l,fm} \cdot I_{l,nc}}{I_{l,fm} + I_{l,nc}} \tag{A9}$$

Assuming a much slower condensation of $H_2SO_4$ than its production leads to an equilibrium state in which the total condensation rate equals the production rate of $H_2SO_4$:

$$\tilde{Co}_3 = \sum_l \tilde{Co}_{3,l} = \dot{M}_3 \tag{A10}$$

with $\dot{M}_3$ the third moment production rate of gaseous $H_2SO_4$. A third-moment condensation rate can be formulated as

$$\tilde{Co}_{3,l} = \sum_l \tilde{Co}_{3,l} = \dot{M}_3 \Omega_l \tag{A11}$$

using a dimensionless coefficient defined as:

$$\Omega_l = \frac{\tilde{Co}_{3,l}}{\tilde{Co}_3} = \frac{\tilde{Co}_{3,l}}{\dot{M}_3} = \frac{I_l}{\sum_l I_l} \tag{A12}$$

Thus, the condensation rate is independent from $\chi_T$ now and can be written for the third moment as:

$$Co_{3,l} = \frac{\pi}{6} \frac{\rho_{H2SO4}}{\rho_a} \dot{M}_3 \Omega_l \tag{A13}$$

In ICON-ART, we approximate $\dot{M}_3$ by the $H_2SO_4$ mass mixing ratio and the model time step. Thus, for Eq. (A13) it follows:

$$Co_{3,l} = \frac{c_{H2SO4}}{\Delta t} \Omega_l \tag{A14}$$

## A3   Coagulation

The parametrizations for the coagulation follow Riemer (2002) and references therein (mainly Whitby et al. (1991)). The rates for the zeroth and third moment are given by

$$Ca_{0,i} = Ca_{0,ii} + Ca_{0,ij} \tag{A15}$$

and

$$Ca_{3,i} = Ca_{3,ij} \tag{A16}$$

with the $Ca_{k,ii}$ and $Ca_{k,ij}$ for the intra- and inter-modal coagulation rates of the $k$-th moment, respectively, and $i$ and $j$ referring to two different modes. In ICON-ART, only coagulation due to Brownian motion is considered so far and reads for a system of two modes $i$ and $j$ as (Whitby et al., 1991)

$$\tilde{Ca}_{0,ii} = \frac{1}{2} \int_0^\infty \int_0^\infty \beta(d_1, d_2) \psi_{0,i}(d_1) \psi_{0,i}(d_2) \mathrm{d}d_1 \mathrm{d}d_2, \tag{A17}$$



$$\tilde{C}a_{0,ij} = \int\limits_0^\infty \int\limits_0^\infty \beta(d_1, d_2)\psi_{0,i}(d_1)\psi_{0,j}(d_2)\mathrm{d}d_1\mathrm{d}d_2, \tag{A18}$$

and

$$\tilde{C}a_{3,ij} = \int\limits_0^\infty \int\limits_0^\infty d_1^3 \beta(d_1, d_2)\psi_{0,i}(d_1)\psi_{0,j}(d_2)\mathrm{d}d_1\mathrm{d}d_2. \tag{A19}$$

Usually only numerical solutions exist for $\beta$ for all sizes, but analytical solutions for certain size regimes exist. The method by Whitby et al. (1991) uses analytic solutions for the integrals in Eq. (A17) to (A19) for the near-continuum and free-molecular regime. The following line only derive the coagulation rates for inter-modal coagulation. The rates for intra-modal coagulation can be derived analogously.

The coagulation coefficient in the near-continuum is given by

$$\beta_{nc}(d_i, D_j) = 2\pi(D_i + D_j)(d_i + d_j). \tag{A20}$$

with $D_l$ the diffusion coefficient of particles in mode $l$:

$$D_l = \frac{k_b T C_l}{3\pi\mu d_l} \tag{A21}$$

Here, $k_B$ denotes the Boltzmann constant, $T$ the temperature, $\mu$ the dynamic viscosity of air, $C_l$ the Cunningham correction factor, and the particle diameter $d_l$. The Cunningham factor $C_l$ is approximated by

$$C_l \approx 1 + 1.246 Kn_l \tag{A22}$$

with $Kn_l = \frac{2\lambda_{air}}{d_l}$ the Knudsen number and $\lambda_l$ the mean free path of air. The coagulation rate in the near continuum regime can now be written as

$$
\begin{aligned}
\tilde{C}a_{0,ij} &= \int\limits_0^\infty \int\limits_0^\infty \beta_{nc}(d_1, d_2)\psi_{0,i}(d_1)\psi_{0,j}(d_2)\mathrm{d}d_1\mathrm{d}d_2 \\
&= \Psi_{0,i}\Psi_{0,j}\rho_a^2 K_{nc} d_{0,i}^3 \left[ 2e^{\frac{36}{8}ln^2(\sigma_i)} + a_i Kn_{gi}\left( e^{\frac{16}{8}ln^2(\sigma_i)} + \frac{d_{0,j}}{d_{0,i}}e^{\frac{4}{8}ln^2(\sigma_i)}e^{\frac{4}{8}ln^2(\sigma_j)} \right) \right. \\
&\left. + a_j Kn_{gj}\left( e^{\frac{36}{8}ln^2(\sigma_j)}e^{\frac{4}{8}ln^2(\sigma_j)} + \frac{d_{0,i}}{d_{0,j}}e^{\frac{64}{8}ln^2(\sigma_j)}e^{\frac{16}{8}ln^2(\sigma_i)} \right) + \left( \frac{d_{0,i}}{d_{0,j}} + \frac{d_{0,j}}{d_{0,i}} \right)e^{\frac{4}{8}ln^2(\sigma_j)}e^{\frac{4}{8}ln^2(\sigma_i)} \right]
\end{aligned}
\tag{A23}
$$

with the factor $K_{nc} = \frac{2k_B T}{3\mu}$. Analogously, the intr-modal coagulation rates $\tilde{C}a_{0,ii}^{nc}$ and $\tilde{C}a_{0,jj}^{nc}$ are derived. The coagulation

rate for the third moment in the near continuum regime reads as:

$$
\begin{aligned}
\tilde{C}a_{3,ij} &= \int\limits_0^\infty \int\limits_0^\infty d_i^3 \beta_{nc}(d_1, d_2)\psi_{0,i}(d_1)\psi_{0,j}(d_2)\mathrm{d}d_1\mathrm{d}d_2 \\
&= \Psi_{0,i}\Psi_{0,j}\rho_a^2 K_{nc}\left[ 2 + a_i Kn_{gi}\left( e^{\frac{4}{8}ln^2(\sigma_i)} + \frac{d_{0,j}}{d_{0,i}}e^{\frac{16}{8}ln^2(\sigma_i)}e^{\frac{4}{8}ln^2(\sigma_j)} \right) \right. \\
&\left. + a_j Kn_{gj}\left( e^{\frac{4}{8}ln^2(\sigma_j)} + \frac{d_{0,i}}{d_{0,j}}e^{\frac{16}{8}ln^2(\sigma_j)}e^{\frac{4}{8}ln^2(\sigma_i)} \right) + \frac{d_{0,j}}{d_{0,i}}e^{\frac{16}{8}ln^2(\sigma_j)}e^{\frac{4}{8}ln^2(\sigma_i)} + \frac{d_{0,i}}{d_{0,j}}e^{\frac{64}{8}ln^2(\sigma_j)}e^{\frac{4}{8}ln^2(\sigma_i)} \right]
\end{aligned}
\tag{A24}
$$





For the free -molecular regime the approximation of the coagulation coefficient is given by

$$\beta_{fm}(d_i, d_j) = \sqrt{\frac{6k_B T}{\rho_{p,i} + \rho_{p,j}}} \left( \sqrt{d_i} + 2\frac{d_j}{\sqrt{d_i}} + \frac{d_j^2}{d_i^{3/2}} + \frac{d_i^2}{d_j^{3/2}} + 2\frac{d_i}{\sqrt{d_j}} + \sqrt{d_j} \right). \tag{A25}$$

The resulting coagulation rate in the free-molecular regime becomes for the zeroth moment

$$
\begin{aligned}
\tilde{C}a_{0,ij}^{fm} &= \int_0^\infty \int_0^\infty \beta_{fm}(d_1, d_2)\psi_{0,i}(d_1)\psi_{0,j}(d_2)\mathrm{d}d_1\mathrm{d}d_2 \\
&= \Psi_{0,i}\Psi_{0,j}\rho_a^2 \sqrt{\frac{6k_B T}{\rho_{p,i} + \rho_{p,j}}} b_0 \sqrt{d_{0,i}} \left[ e^{\frac{1}{8}ln^2(\sigma_i)} + \sqrt{\frac{d_{0,j}}{d_{0,i}}} e^{\frac{1}{8}ln^2(\sigma_j)} + 2\frac{d_{0,j}}{d_{0,i}} e^{\frac{1}{8}ln^2(\sigma_i)} e^{\frac{4}{8}ln^2(\sigma_j)} \right. \\
&\quad \left. + \frac{d_{0,j}^2}{d_{0,i}^2} e^{\frac{9}{8}ln^2(\sigma_i)} e^{\frac{16}{8}ln^2(\sigma_j)} + \left( \sqrt{\frac{d_{0,i}}{d_{0,j}}} \right)^3 e^{\frac{16}{8}ln^2(\sigma_i)} e^{\frac{9}{8}ln^2(\sigma_j)} + 2\sqrt{\frac{d_{0,i}}{d_{0,j}}} e^{\frac{4}{8}ln^2(\sigma_i)} e^{\frac{1}{8}ln^2(\sigma_j)} \right]
\end{aligned}
\tag{A26}
$$

and for the third moment

$$
\begin{aligned}
\tilde{C}a_{0,ij}^{fm} &= \int_0^\infty \int_0^\infty d_i^3 \beta_{fm}(d_1, d_2)\psi_{0,i}(d_1)\psi_{0,j}(d_2)\mathrm{d}d_1\mathrm{d}d_2 \\
&= \Psi_{0,i}\Psi_{0,j}\rho_a^2 \sqrt{\frac{6k_B T}{\rho_{p,i} + \rho_{p,j}}} b_3 (d_{0,i})^{\frac{7}{2}} \left[ e^{\frac{49}{8}ln^2(\sigma_i)} + \sqrt{\frac{d_{0,j}}{d_{0,i}}} e^{\frac{36}{8}ln^2(\sigma_i)} e^{\frac{1}{8}ln^2(\sigma_j)} + 2\frac{d_{0,j}}{d_{0,i}} e^{\frac{25}{8}ln^2(\sigma_i)} e^{\frac{4}{8}ln^2(\sigma_j)} \right. \\
&\quad \left. + \frac{d_{0,j}^2}{d_{0,i}^2} e^{\frac{9}{8}ln^2(\sigma_i)} e^{\frac{16}{8}ln^2(\sigma_j)} + \left( \sqrt{\frac{d_{0,i}}{d_{0,j}}} \right)^3 e^{\frac{100}{8}ln^2(\sigma_i)} e^{\frac{9}{8}ln^2(\sigma_j)} + 2\sqrt{\frac{d_{0,i}}{d_{0,j}}} e^{\frac{64}{8}ln^2(\sigma_i)} e^{\frac{1}{8}ln^2(\sigma_j)} \right].
\end{aligned}
\tag{A27}
$$

The values for $b_0$ and $b_3$ are given by Whitby et al. (1991) with $b_0 = 0.9$ and $b_3 = 0.9$ for inter-modal coagulation and $b_0 = 0.8$ for intramodal coagulation.

Finally a harmonic mean is applied, similar to the procedure for the condensation, to receive the coagulation rate for the full size range. For the conversion to coagulation rate of the number concentration and mass mixing ratio, the density of air and mode density are considered:

$$Ca_{0,l} = \frac{1}{\rho_a} \frac{\tilde{C}a_{0,ij}^{nc} \cdot \tilde{C}a_{0,ij}^{fm}}{\tilde{C}a_{0,ij}^{nc} + \tilde{C}a_{0,ij}^{fm}} \tag{A28}$$

and

$$Ca_{3,l} = \frac{\rho_{p,l}}{\rho_a} \frac{\tilde{C}a_{3,ij}^{nc} \cdot \tilde{C}a_{3,ij}^{fm}}{\tilde{C}a_{3,ij}^{nc} + \tilde{C}a_{3,ij}^{fm}} \tag{A29}$$

## Appendix B: Database of prescribed aerosol optics

To represent the optical properties of aerosols in the model, we use a set of predefined modes that account for typical particle size distributions and compositions. These modes are based on lognormal distributions, described by their count median





diameter (cmd) and geometric standard deviation ($\sigma$g). In addition to the size distribution, each mode specifies the chemical
composition of the particle core and, if present, a coating layer. The coating fraction is defined as the ratio of the coating
thickness to the core radius. The data are provided in the model repository and summarized in Table B1

**Table B1.** List of available modes, the corresponding parameters of the particle size distribution and the particle composition, for whitch
the optical properties are precalculated. The assumed particle size distribution is characterized by the count median diameter $cmd$ and the
geometric standard deviation $\sigma_g$. 'core' and 'coating' describe the constituents of the particles' core and its coating, if one is present. The
coating fraction (coat. frac.) is the ratio of the thickness of the coating shell and the radius of the core. Refractive indices are taken from
Gordon et al. (2022). Data for aerosols with non-sphericity approximation are taken from Hoshyaripour et al. (2019).

| name | $cmd$ [nm] | $\sigma_g$ | core | coating | coat. frac. | description |
|---|---|---|---|---|---|---|
| inorg_ait | 10 | 1.70 | inorganic | | | sulfate in Aitken mode |
| inorg_acc | 200 | 2.00 | inorganic | | | sulfate in accumulation mode |
| inorg_coa | 2000 | 2.20 | inorganic | | | sulfate in coarse mode |
| ash_acc | 200 | 2.00 | ash | | | basaltic ash in accumulation mode |
| ash_coa | 2000 | 2.20 | ash | | | basaltic ash in coarse mode |
| ash_giant | 12000 | 2.00 | ash | | | giant ash particles |
| ash_a_mie | 1190 | 1.41 | ash | | | ash of mode A |
| ash_b_mie | 2320 | 1.60 | ash | | | ash of mode B |
| ash_c_mie | 3920 | 1.76 | ash | | | ash of mode C |
| dust_a | 644 | 1.70 | dust | | | dust of mode A with non-sphericity approximation |
| dust_b | 3454 | 1.60 | dust | | | dust of mode B with non-sphericity approximation |
| dust_c | 8672 | 1.50 | dust | | | dust of mode C with non-sphericity approximation |
| dust_a_mie | 644 | 1.70 | dust | | | dust of mode A |
| dust_b_mie | 3454 | 1.60 | dust | | | dust of mode B |
| dust_c_mie | 8672 | 1.50 | dust | | | dust of mode C |
| seasalt_a_mie | 100 | 1.90 | seasalt | | | seasalt of mode A |
| seasalt_b_mie | 3000 | 2.00 | seasalt | | | seasalt of mode B |
| seasalt_c_mie | 30000 | 1.70 | seasalt | | | seasalt of mode C |
| soot_ait | 20 | 1.70 | soot | | | soot (OC/BC=30) in Aitken mode |
| soot_acc | 150 | 2.00 | soot | | | soot (OC/BC=30) in accumulation mode |
| soot_dwd | 150 | 2.00 | soot | | | coated soot (OC/BC=30) used for weather forecast by DWD |
| icoat15_ash_acc | 200 | 2.00 | ash | inorganic | 0.15 | ash with sulfate coating in accumulation mode |
| icoat15_ash_coa | 2000 | 2.20 | ash | inorganic | 0.15 | ash with sulfate coating in coarse mode |
| ocoat15_ash_acc | 200 | 2.00 | ash | organic | 0.15 | ash with SOA coating in accumulation mode |
| ocoat15_ash_coa | 2000 | 2.20 | ash | organic | 0.15 | ash with SOA coating in coarse mode |
| icoat15_dust_acc | 200 | 2.00 | dust | inorganic | 0.15 | dust with sulfate coating in accumulation mode |
| icoat15_dust_coa | 2000 | 2.20 | dust | inorganic | 0.15 | dust with sulfate coating in coarse mode |
| ocoat15_dust_acc | 200 | 2.00 | dust | organic | 0.15 | dust with SOA coating in accumulation mode |
| ocoat15_dust_coa | 2000 | 2.20 | dust | organic | 0.15 | dust with SOA coating in coarse mode |
| icoat15_soot_ait | 20 | 1.70 | soot | inorganic | 0.15 | soot (OC/BC=30) with sulfate coating in Aitken mode |
| icoat15_soot_acc | 150 | 2.00 | soot | inorganic | 0.15 | soot (OC/BC=30) with sulfate coating in accumulation mode |
| ocoat15_soot_ait | 20 | 1.70 | soot | organic | 0.15 | soot (OC/BC=30) with SOA coating in Aitken mode |
| ocoat15_soot_acc | 150 | 2.00 | soot | organic | 0.15 | soot (OC/BC=30) with SOA coating in accumulation mode |



## Appendix C: Preprocessing for MECCA chemistry simulations

Figure 5 illustrates a schematic of the preprocessing steps required to integrate the MOZART-4 chemistry mechanism into ART. The steps are as follows: Within MECCA (`caaba_4.0/mecca`), the `eqn` folder contains various chemical reaction mechanisms formatted for KPP, with the corresponding gas-phase species defined in `gas.spc`. To activate MOZART-4 chemistry, the `xmecca` script must be executed using the `mozart.bat` batch file, which utilizes the KPP equation file `eqn/mozart/mozart.eqn`. The batch file also allows users to select a subset of the full mechanism—for example, enabling all gas-phase reactions while excluding those involving halogens. Based on the selected mechanism (`mecca.eqn`) and species list (`mecca.spc`), KPP generates Fortran90 files (`messy_mecca_kpp_*.f90`) for the numerical integration of the ODEs derived from the reaction mechanism.

The MECCA preprocessing has been extended to couple KPP routines from the box model to ART. A shell script (`create_icon_code.sh`), along with an AWK-based routine (`write_tracers_to_xml.awk`), generates the necessary files and copies them to the corresponding ART directory. Additionally, the tracer XML file `mecca_tracers.xml` is created. This process uses the table `caaba_4.0/mecca/process_gas.tbl` to define advection properties: if advection is enabled, the advection template `hadv52aero` is selected; otherwise, the `<transport>` tag is set to `off`. Users can modify or extend this table to accommodate their specific mechanism and setup. The molecular weight of the tracers is computed based on the chemical formulas provided in `gas.eqn`. The scripts for these steps are provided in the supplementary material.

To integrate the MOZART-T1 mechanism into ART, the following steps should be followed. Using the reactions and rate coefficients provided in the supplementary material of Emmons et al. (2020), users must create an equation file, `mozart-t1.eqn`, containing the relevant chemical reactions. The syntax for this file is outlined in the CAABA/MECCA user manual available in the manual directory of `caaba_4.0`. Additionally, the species file `gas.spc` must be updated accordingly. For convenience, both `mozart-t1.eqn` and the modified `gas.spc` file for MOZART-T1 are provided in the supplementary material.

Next, the batch file `mozart.bat` must be adjusted to select `mozart-t1.eqn` as the active mechanism. The required KPP files can then be generated as described earlier. Finally, to define the advection properties of tracers in the XML file, users need to extend the table `caaba_4.0/mecca/process_gas.tbl` by including all MOZART-T1 species with their corresponding advection properties.

## Appendix D: XML tags and namelist settings for the Online Emissions Module

The Online Emissions Module (OEM) has its own namelist section oemcntrl_nml, where the paths to the different input files (in netcdf format) need to be defined. Table D1 lists all possible entries. The last column describes in which case the file is required. The file hour_of_year_nc, for example, is only required if for one of the tracers in the tracer XML file the value of tscale is set to 2.





**Table D1.** List of all oemctrl_nml namelist parameters controlling the input for OEM.

| Parameter | Description | Restriction |
|---|---|---|
| gridded_emissions_nc | path to the file with gridded emissions | |
| vertical_profile_nc | path to the file with vertical profiles | |
| hour_of_day_nc | path to the file with temporal profiles for hour_of_day | tscale=1 |
| day_of_week_nc | path to the file with temporal profiles for day_of_week | tscale=1 |
| month_of_year_nc | path to the file with temporal profiles for month_of_year | tscale=1 |
| hour_of_year_nc | path to the file with temporal profiles for hour_of_year | tscale=2 |
| ens_reg_nc | path to the region mask for ensemble generation | oem_type=ens |
| ens_lambda_nc | path to the scaling factors for ensemble generation | oem_type=ens |
| boundary_regions_nc | path to the region mask for latbc perturbation | use of oem_bg_ens |
| boundary_lambda_nc | path to the scaling factors for latbc perturbation | use of oem_bg_ens |

**Table D2.** List of all XML tags available to control the OEM tracers.

| Tag | Type | Restriction | Description | Remarks |
|---|---|---|---|---|
| oem_type | "char" | emis,ens | defines the type of OEM tracer | main tag |
| oem_cat | "char" | | comma-separated list of category names for emissions | |
| oem_tscale | "int" | 1,2 | type of temporal profiles | |
| oem_tp | "char" | | comma-separated list of category names for temporal profiles | |
| oem_vp | "char" | | comma-separated list of category names for vertical profiles | |
| oem_bg_ens | "char" | oem_type=ens | name of background tracer that is added to the ensemble member and perturbed | |

In order to prepare a tracer for OEM, a number of tags need to be defined in the tracer XML file. An overview of all OEM tags is presented in Tab. D2. In order to be able to add emissions to a tracer using OEM, the tag oem_type needs to be set for this tracer to "emis". The tag oem_cat lists all the emission categories, which contribute to emissions of this tracer. The tags oem_tp and oem_vp need to be lists of the same length as oem_cat and provide the names of the temporal and vertical profiles that should be used for each emission category. Temporal profiles can either be composed of the product of hour-of-day, day-of-week and month-of-year profiles (if tag oem_tscale=1) or of a single profile with scaling factors for each hour-of-year (oem_tscale=2).





## Appendix E: XML tags for deposition of gases

### E1 Dry deposition

The dry deposition scheme is available for both parameterized and full chemistry simulations via the tracer XML file by setting the following tags. To activate dry deposition for a specific tracer, one needs to set:

```
<idep type="int">1</idep>
```

Additionally, the reactivity of the tracer must be specified using the tag `<reac>`, which should be a value between 0 and 1. The solubility of the tracer is defined by the tag `<heff>`, which represents its effective Henry's law constant in units of M atm⁻¹.

To compute the resistances representing diffusive processes, the molecular diffusivity of the species must also be provided. The user can specify two values for this: `<vdmol1>` and `<vdmol2>`. The tag `<vdmol1>` specifies the ratio $D_{\text{H}_2\text{O}}/D_i$ of the molecular diffusivities of H$_2$O and the specific tracer, while `<vdmol2>` encodes the ratio from Equation (9), more precisely $\texttt{vdmol2} = 1/f_D$.[1]

### E2 Wet deposition

To activate wet deposition for a given tracer, the following tag must be set in the tracer XML file:

```
<iwash type="int">1</iwash>
```

The scavenging ratios for in-cloud and below-cloud scavenging are specified with the tags `<win>` and `<wsub>`, respectively.

## Appendix F: Lifetime-based tracers

In the following equations, $k$ depicts the destruction rates for the given tracer, $P$ the pressure in Pascal, COL the column of the
given gas, and $z_p$ the pressure altitude in m.

– TRCH4: parameterization by ECMWF ported to ICON-ART (Schröter et al., 2018)

$$
k_{CH_4}(P) = \begin{cases} \frac{1}{86400 \times 100} & \text{if } P \leq 50 \\[3mm] \frac{1}{86400 \times \left(100 \times \left(1 + \frac{19\ln(10)}{(\ln(20))^4} \frac{(\ln(P/50))^4}{\ln(10000/P)}\right)\right)} & \text{if } 50 < P < 10000 \\[3mm] 0 & \text{if } P \geq 10000 \end{cases}
\tag{F1}
$$

---

[1]In the current version of the dry deposition routine, $D_0 = D_{\text{H}_2\text{O}}$, so vdmol1 = vdmol2.





- TRCH3CN: parameterization as described in Fischbeck (2018) and based on Harrison and Bernath (2013) and Singh et al. (2003):

$$k_{CH_3CN} = \begin{cases} 1/2207520000 & \text{above tropopause} \\ 1/37843200 & \text{below tropopause} \\ 1/1814400 & \text{in the lowest nine levels above the ocean} \end{cases} \tag{F2}$$

- TRCO2: parameterization from (Schröter et al., 2018)

$$k_{CO_2} = 2.20 \times 10^{-8} \exp(-10^{-20} COL_{O_2}) \tag{F3}$$

- TRCO: parameterization from (Schröter et al., 2018)

$$k_{CO}(z_p) = \frac{2.0 \times 10^{-8}}{1 + \exp\left(\frac{z_p - 85000}{2800}\right)} + 10^{-7} \exp\left(-\frac{(z_p - 70000)^2}{7000^2}\right) + 2 \times 10^{-6} \exp\left(-\frac{(z_p - 45000)^2}{14000^2}\right) \tag{F4}$$

- TRH2O: parameterization based on Brasseur and Solomon (2006)

$$k_{H_2O}(P) = \begin{cases} \frac{1}{86400 \times 3} & \text{if } P \leq 0.1 \\ \\ \frac{1}{86400\left(\frac{1}{\exp}\left(\log\left(\frac{1}{3} + 0.01\right) - 0.5\left(\log(100) + \log\left(\frac{1}{3} + 0.01\right)\left(1 + \cos\left(\pi \log\left(\frac{P}{20}\right)/\log(0.005)\right)\right)\right)\right) - 0.01\right)} & \text{if } 0.1 < P < 20 \\ \\ 0 & \text{if } P \geq 20 \end{cases}$$

$$\tag{F5}$$