# Peer review of "The Atmospheric Composition Component of the ICON modeling framework: ICON-ART version 2025.04"

_EGUsphere, 2025_

## Referee Comment (RC2)

**Review report of Hoshyaripour et al. 2025**

In this manuscript, the authors present a comprehensive description of the latest developments in the ART atmospheric model as coupled to the ICON meteorological model (ICON-ART v2025.04). They have pitched this model as a novel tool which allows for a unified simulation of atmospheric composition and climate across scales without compromising on any key aspects of atmospheric chemistry and physics. That is, it represents key processes such as atmospheric chemistry for trace gases and aerosols, aerosol growth dynamics, and their interaction with meteorology via radiative transfer, aerosol-radiation interactions, and aerosol cloud interactions allowing for atmospheric composition-climate feedback simulations at varying scales within a single global framework. This, as I understand it, eliminates the need for running separate limited-area models (LAMs) that depend on lateral boundary conditions from coarse global models which often introduces abrupt and unrealistic meteorological features within the simulations and also lead to inconsistencies in the treatment of physical and chemical processes.

The authors have done an excellent job of detailing all key modules within the modelling framework, from emission processes and sources, chemistry schemes which vary from simplistic to sophisticated, deposition schemes, aerosol sources ranging from natural to anthropogenic, multiple mixing states, aerosol dynamics, aerosol-radiation interactions (which now incorporate a novel ML-based approach which better treats aerosol mixing states and results in a more accurate radiation perturbation), and aerosol-cloud interactions. They have also described multiple ways in which the aerosol-radiation interactions are called within the model which allows the user to study the direct vs holistic/interactive impacts of aerosols of different sizes on climate variables.

They have also provided a fairly clear description of the code infrastructure and the coupling between ICON and ART models (although such things remain largely opaque to basic model users unless they're willing to do the due diligence of probing into the code themselves, but it's a good starting point). The authors have also discussed the portability of ICON-ART on GPUs for potential speeding-up of simulations, which is a work in progress; it's good to see this forward-thinking. Standard configurations are also discussed which show the model's flexibility for various applications where there's more focus on certain processes and less on others.

Overall, I have to say this is very impressive work - both the actual development of the model and its clear documentation in this manuscript. Therefore my comments are fairly minor. I have mentioned them pointwise below:

L18: "essential for improving predictions related to weather, renewable energy, climate change, air pollution..." consider changing to:

"essential for improving predictions **and understanding** related to weather, renewable energy, climate change, air pollution..."

L59-60: "OEM enables efficient processing of emissions that are constant in time or changing only temporally, but not spatially"

This is not clear to me (and may also bother other readers): if different sources are varying differently temporally, it means emissions overall are changing spatially too - please clarify this.

While the paper excels at describing what has been implemented, it could be strengthened by briefly showing why some of the new developments matter in a more quantitative sense. For example:

Section 2.4 (Sea Salt): The new Grythe et al. (2014) parameterization is included to better represent SST dependence. A simple zonal-mean plot or a brief statement quantifying the typical change in sea salt emissions or burden in tropical regions compared to the older scheme would be highly illustrative.

Section 4.4 (Subpollen Particles): The parameterization for SPP release is described. It would be beneficial to include a sentence stating the typical order-of-magnitude contribution of SPPs to total aerosol number concentration or CCN in relevant regions during pollen season, even if citing another study (Werchner et al., 2022?).

Section 3.2 (Detailed chemistry mechanisms): a brief comment on the typical computational cost increase when moving from a simplified chemistry scheme (like Linoz) to a full mechanism (like MOZART-T1) would provide valuable context for users planning simulations. A percentage increase in runtime, similar to that provided for LINOZ in Section 3.1.3, would be sufficient.

Figure 6: The inner and outer circles are unclear (visually and also in terms of values). For the winter plot (left), the outer rings mostly match the surrounding areas on the contour map but for the summer plot (right) these outer rings are consistently of a lighter shade than the surrounding values in the map which suggests some issue with sampling - please double check. If I disregard the outer rings and only compare the inner circle values with the surrounding values on the simulated map, I see a better model-obs agreement. However, when comparing the inner circles with outer rings, it looks like the model is underestimating surface ozone in both winter and summer. This underestimation doesn't sit well with the broader context of basically all global and regional models overestimating Northern Hemispheric surface ozone (e.g., Young et al., 2013; 2018; Ansari et al., 2025; Nalam et al., 2025, Gao et al., 2025). I suggest that the authors make this figure simpler by only showing one solid circle representing only observed values, and include the overall mean bias, RMSE, and correlation coefficient r for both seasons somewhere in the figure and the text. Accordingly, the text that "the model accurately reproduces..." should be made more nuanced and discussed in the broader context of the

aforementioned papers. The authors must also mention which emission inventory was used for these LAM simulations over Europe. The authors should discuss potential reasons for O3 underestimation.

L432: "hats and overbars"?

L491: "processes such as removal processes" to "processes such as removal mechanisms"?

L493: Describe the key aspects of this alternative method in a couple of sentences here, especially in relation to its computational efficiency.

L571: "implemented in other models": name those models here along with the citations.

L669 (or thereabouts): Also include a couple of sentences on the best practices of using this dusty cirrus parameterization for different (coarser, finer, or variable) grid resolutions. How does it perform across scales? Has this been tested? This could be discussed a bit.

Figure 11: The average OLR value should be shown in enlarged font or ideally printed over the map, or the reader might miss it. The technical name of the simulation experiment is not needed on the figure.

Figure 12: This schematic could be improved: include additional boxes at the top showing input data (for both ICON and ART). Name some typical variables (winds, moisture, pressure; anthro emissions). Similarly, name some typical output variables from ICON and ART; add additional boxes if necessary. Use appropriate arrows along the lines to indicate the direction of control and sequence of execution and data flow. Aim to better depict the loops and subloops within the model time integration workflow. In the caption, consider changing "circles" to "loops".

L698: "since also diagnostic variables can be defined with it" to "since diagnostic variables can also be defined with it".

This manuscript is an exemplary model description paper. It is extremely well-written and thoroughly documents a critical tool for the atmospheric modelling community. The suggested revisions are minor and are aimed at making an already excellent paper even better. I strongly recommend its publication in GMD once these comments are addressed.

**References:**

Ansari, T., Nalam, A., Lupaşcu, A., Hinz, C., Grasse, S., and Butler, T.: Explaining trends and changing seasonal cycles of surface ozone in North America and Europe over the 2000–2018 period: A global modelling study with NOx and VOC tagging, EGUsphere [preprint], https://doi.org/10.5194/egusphere-2024-3752, 2024.

Gao, Y., Kou, W., Cheng, W., Guo, X., Qu, B., Wu, Y., et al. (2025). Reducing long-standing surface ozone overestimation in Earth system modeling by high-resolution simulation and dry deposition improvement. Journal of Advances in Modeling Earth Systems, 17, e2023MS004192. https://doi.org/10.1029/2023MS004192

Nalam, A., Lupascu, A., Ansari, T., and Butler, T.: Regional and sectoral contributions of NOx and reactive carbon emission sources to global trends in tropospheric ozone during the 2000–2018 period, EGUsphere [preprint], https://doi.org/10.5194/egusphere-2024-432, 2024.

Young, P. J., Archibald, A. T., Bowman, K. W., Lamarque, J.-F., Naik, V., Stevenson, D. S., Tilmes, S., Voulgarakis, A., Wild, O., Bergmann, D., Cameron-Smith, P., Cionni, I., Collins, W. J., Dalsøren, S. B., Doherty, R. M., Eyring, V., Faluvegi, G., Horowitz, L. W., Josse, B., Lee, Y. H., MacKenzie, I. A., Nagashima, T., Plummer, D. A., Righi, M., Rumbold, S. T., Skeie, R. B., Shindell, D. T., Strode, S. A., Sudo, K., Szopa, S., and Zeng, G.: Pre-industrial to end 21st century projections of tropospheric ozone from the Atmospheric Chemistry and Climate Model Intercomparison Project (ACCMIP), Atmos. Chem. Phys., 13, 2063–2090, https://doi.org/10.5194/acp-13-2063-2013, 2013.

P. J. Young, V. Naik, A. M. Fiore, A. Gaudel, J. Guo, M. Y. Lin, J. L. Neu, D. D. Parrish, H. E. Rieder, J. L. Schnell, S. Tilmes, O. Wild, L. Zhang, J. Ziemke, J. Brandt, A. Delcloo, R. M. Doherty, C. Geels, M. I. Hegglin, L. Hu, U. Im, R. Kumar, A. Luhar, L. Murray, D. Plummer, J. Rodriguez, A. Saiz-Lopez, M. G. Schultz, M. T. Woodhouse, G. Zeng; Tropospheric Ozone Assessment Report: Assessment of global-scale model performance for global and regional ozone distributions, variability, and trends. Elementa: Science of the Anthropocene 1 January 2018; 6 10. doi: https://doi.org/10.1525/elementa.265

---

## Author Comment (AC3)

**Response to Reviewers**

**Manuscript: Hoshyaripour et al. (2025), GMD**

We sincerely thank both reviewers for their detailed and constructive comments, which helped us improve the manuscript.

Reviewer comments are presented in blue, our replies follow in black, followed by the corresponding changes in the revised manuscript in italic format.

**Reviewer 1:**

- 1. The large number of the acronyms has been used in the whole manuscript and it is suggested to add a separate Appendix describing this manuscript.
- We added an Appendix for clarity of Acronyms.

**Appendix A: Acronyms**

| ICON | ICOsahedral Non-hydrostatic Aerosols and Reactive Trace gases |  |
|------|---------------------------------------------------------------|--|
| ART  |                                                               |  |
| OEM  | Online Emission Module                                        |  |

**VPRM** Vegetation Photosynthesis and Respiration Model

ARI Aerosol-Radiation Interaction
ACI Aerosol-Cloud Interaction

**CAMx** Comprehensive Air Quality Model with Extensions

CAABA Chemistry As A Boxmodel Application

MECCA Module Efficiently Calculating the Chemistry of the Atmosphere

KPP Kinetic Pre-Processor FKB Fortran-Keras Bridge LAM Limited-Area Mode

**MOZART** Model for Ozone and Related chemical Tracers

**LINOZ** LINearized OZone

**NMVOC** Non-Methane Volatile Organic Compounds

**NO**x Nitrogen Oxides

NOy Reactive Nitrogen Compounds PSCs Polar Stratospheric Clouds

**NAT** Nitric Acid Trihydrate

STS Supercooled Ternary Solution
INAS Ice Nucleation Active Site
SPPs Subpollen Particles
DRE Direct Radiative Effect

**EDGAR** Emission Database for Global Atmospheric Research

CAMS-REG Copernicus Atmosphere Monitoring Service Regional inventory

GNFR Gridded Nomenclature For Reporting
GFAS Global Fire Assimilation System

**FRP** Fire Radiative Power

- 2. The caption of the Table and Figure could be modified to be self-explanatory. e.g., Table 1-2 give the brief overview of the Basis v Implementation. It can re-written what basis is about etc.
- Both tables 1 and 2 are revised by providing additional info in the caption.

Table 1: Emissions in the ICON-ART model system including the main technical/scientific basis and references of the parameterization and the first published implementation in the ART framework.

Table 2: Types of chemistry in ART including reference to their main technical/scientific descriptions and the first published implementation in the ICON-ART framework.

- 3. As noted in lines 43–45, previous work with ICON-ART has been acknowledged, while this manuscript aims to present an updated overview. However, it would be helpful to clarify which components are entirely new in the version 25.04 and which represent updates to existing implementations. For instance, while the Online Emission Model (OEM) is mentioned as part of version 25.04, the manuscript does not clearly indicate how this anthropogenic emission component was handled in earlier versions. In contrast, Section 2.3, which covers volcanic eruptions, provides an excellent and detailed account of the updates made—offering a useful model for how other sections might be strengthened with similar clarity.
- Our intention in Table 1 and the accompanying text was to highlight only the new or substantially revised features introduced in this paper, while referring readers to the existing model description papers for all previously implemented components. To make this clearer, we have now highlighted in Table 1 the implementations that are new and represent updates of previously published implementations.

Table 1: Emissions in the ICON-ART model system including the main technical/scientific basis and references of the parameterization and the first published implementation in the ART framework. The new features presented in this work are shown in bold.

| Emission Type | Technical Basis & Reference(s)            | Implementation in ART                         |
|---------------|-------------------------------------------|-----------------------------------------------|
| Anthropogenic | Prescribed (Weimer et al., 2017), OEM     | Weimer et al. (2017); Jähn et al. (2020), see |
|               | (Jähn et al., 2020)                       | Sect. 2.1                                     |
| Wildfires     | GFAS (Kaiser et al., 2012) and Plume-     | Walter et al. (2016), see Sect. 2.2           |
|               | rise model (Freitas et al., 2007)         |                                               |
| Volcanic      | 1D model FPlume (Folch et al., 2016)      | Bruckert et al. (2022), see Sect. 2.3         |
| Desert Dust   | Saltation-based (Vogel et al., 2006)      | Rieger et al. (2017)                          |
| Sea Salt      | Wave breaking and whitecap formation      | Lundgren et al. (2013); Rieger et al.         |
|               | (Monahan et al., 1986; Smith and Harri-   | (2015); see Sect. 2.4                         |
|               | son, 1998; Mårtensson et al., 2003), SST- |                                               |
|               | dependant (Grythe et al., 2014)           |                                               |
| DMS           | DMS conc. in ocean (Lana et al., 2011)    | see Sect. 2.5                                 |
| Biogenic VOCs | MEGAN (Guenther et al., 2012)             | Weimer et al. (2017)                          |
| Pollen        | EMPOL (Zink et al., 2013)                 | Zink et al. (2013), see Sect. 2.6             |
| Point source  | Rieger et al. (2015)                      | Rieger et al. (2015)                          |

- 4. Section 5.2 provides a comprehensive description of the CCN activation and its coupling with ICON microphysics for liquid-phase clouds. While the methodological explanation is clear, given that ACI is emphasized in the abstract as a key development, this section could be further strengthened by including a brief quantitative validation or sensitivity analysis demonstrating the realized impact of ACI. The INAS-based treatment for ice-phase ACI is still under development and may be incorporated in future work.
- We have added the following text and figures to the CCN activation. As mentioned in the paper, INAS-based activation is already implemented and available in ART but not yet coupled to the 2-mom scheme of ICON.

Figures 11 and 22 show preliminary results from idealized simulations of a warm bubble. The model setup follows the Weisman-Klemp test case (Weisman and Klemp, 1982). A predefined sea salt concentration of 2 × 10^7 #/kg is uniformly distributed throughout the domain and equally distributed between the accumulation and coarse modes. Figure 11 displays the number of activated cloud condensation nuclei (n\_ccn) in #/kg, accumulated over 640 seconds from the start of the simulation. Sea salt aerosols are activated within the updraft region generated by the warm bubble. Figure 12 illustrates the ratio of activated particles to available sea salt aerosols as a function of vertical velocity, for (a) accumulation and (b) coarse mode. The results indicate that, as expected, a substantial fraction of sea salt in the coarse mode gets activated, whereas only a small portion of sea salt in the accumulation mode undergoes activation. This outcome aligns with Köhler theory, which predicts that larger particles are more likely to be activated due to their lower critical supersaturation.

**Figure 11:** Number of activated sea salt aerosols from an idealized warm bubble simulation based on the Weisman-Klemp test case (Weisman and Klemp, 1982).

**Figure 12:** Ratio of activated particles to available sea salt aerosols as a function of vertical velocity, shown for (a) accumulation mode and (b) coarse mode.

- 5. Emission processes such as desert dust and biogenic VOCs (e.g., VPRM, mentioned later) are not discussed in the manuscript. It is suggested to include a brief discussion at the end of these in Section 2, or alternatively, add short descriptions to Table 1 (e.g., in Basis column) to make their inclusion and treatment clearer.
- To improve clarity, we have added short descriptive phrases for the desert dust and biogenic VOC emission processes in Table 1. These components are part of the established ICON-ART emission suite and are not newly implemented in version 25.04; therefore, they were originally described only briefly. We now additionally clarify this in

Section 2 by noting that these processes remain unchanged in the current model version and are summarized in Table 1 with full details available in the cited model description papers. For the modification of Table 1 please see the previous answer. The following text is added to the paper in section 2:

Processes that are already well established in ICON-ART, such as desert dust and biogenic VOC emissions, remain unchanged in version 25.04 and are therefore only briefly summarized in Table 1, with full details provided in the cited model description papers.

 In addition, we have added a short description of how VPRM is used for the simulation of CO2 and added a reference to Ponomarev et al. (2025), where more details are provided. The text describing VPRM is:

VPRM was introduced to enable the simulation of atmospheric carbon dioxide, which is not only affected by anthropogenic emissions but also by exchange with the biosphere. A first application of VPRM in ICON-ART was demonstrated by Ponomarev et al. (2025).

- 6. Line 28-29 Page2: Repetition in the abbreviation defining, ICON, ART etc.
- Since this is a model description paper, the model names (ICON, ART, and ICON-ART)
  are essential identifiers and should appear clearly in the abstract. We therefore retained
  their definitions in the abstract for clarity and discoverability but removed the repeated
  definitions from the Introduction. The Introduction now refers to the models directly,
  assuming prior definition in the abstract.

The ICON model has been developed and widely used for weather and climate prediction across scales. It solves the 3D non-hydrostatic and compressible Navier–Stokes equations on an icosahedral-triangular grid (Gassmann and Herzog, 2008), facilitating precise predictions across scales (Zängl et al., 2015; Heinze et al., 2017; Giorgetta et al., 2018). The ART module, integrated into the ICON framework, enables comprehensive modeling of atmospheric composition.

- 7. Line 52-53: The reference needed which describe OEM in COSMO-ART.
- This is further provided in Table 1 and section 2.1.
- 8. Line 68: Hermes or HERMES (High-Elective Resolution Modelling Emission System)?
- Corrected to HERMES
- 9. Line 67 Is there anything missing in the line '[e.g.,][]'?
- Corrected to (e.g., Menut et al., 2024; Woo et al., 2012)
- 10. Line 175-180: The new Grythe et al. (2014) sea-salt emission parametrization is introduced in Section 2.4. A brief quantitative or visual comparison with the Monahan scheme could further illustrate the improvement in sea-salt emission estimates.
- To address this, we have expanded Section 2.4 with a short qualitative description of the key conceptual differences between the two parameterizations, focusing on their

treatment of whitecap coverage, particle-size distribution, and the explicit SST dependence introduced in Grythe et al. (2014). A detailed quantitative or visual comparison of emission fluxes would require a comprehensive analysis of the full emission—transport—deposition chain to ensure meaningful interpretation. Such an investigation goes beyond the scope of the present manuscript, which aims primarily to document the model developments and technical implementation. We therefore consider the new qualitative comparison sufficient for the purpose of this paper, while a full evaluation is planned for a dedicated follow-up study currently in preparation.

MMS and G14 sea-salt emission schemes differ not only in their whitecap formulations but also in their treatment of particle-size distributions and SST-dependent scaling (Grythe et al., 2014; Barthel et al., 2019; Li et al., 2024). Barthel et al. (2019) demonstrated that SST corrections can substantially reduce coarse-mode concentrations and may even have a larger impact than switching between source functions. They also found the strongest divergences for particles larger than PM2.5, with SST effects further amplifying these differences. These insights highlight that the structural contrasts between MMS and G14 schemes, particularly the inclusion of SST dependence and the size-resolved flux formulation, can significantly influence emitted mass. While a quantitative evaluation is beyond the scope of this study, this context helps to clarify the expected behavior of the new G14 implementation.

**11. Line 211: Meccatracer?**

• We changed the sentence to:

The most complex tracers in ICON are those participating in chemical reactions described by a coupled system of Ordinary Differential Equations (ODE). These tracers are called meccatracers, because they are solved by the atmospheric chemistry module MECCA as described in Section 3.2.

- 12. Line 424-426: The sedimentation terms in Equations (12)–(13) use inconsistent symbols  $(\Phi \to \Psi)$ . Please ensure consistent notation for the prognostic variable, either using the hat over  $\Psi$  throughout or omitting it consistently.
- This was a typo and is corrected accordingly. We use Ψ consistently through the paper.

$$\frac{\partial \left(\bar{\rho_a}\hat{\Psi}_{0,l}\right)}{\partial t} = -\nabla \cdot \left(\hat{v}\bar{\rho_a}\hat{\Psi}_{0,l}\right) - \nabla \cdot \left(\overline{\rho_a}v''\Psi_{0,l}''\right) - \frac{\partial}{\partial z}\left(v_{\text{sed},0,l}\bar{\rho_a}\hat{\Psi}_{0,l}\right) - Wa_{0,l} - Ca_{0,l} - Nu_{0,l} - Em_{0,l}$$
(12)

$$\frac{\partial \left(\bar{\rho_{a}}\hat{\Psi}_{3,l}\right)}{\partial t} = -\nabla \cdot \left(\hat{v}\bar{\rho_{a}}\hat{\Psi}_{3,l}\right) - \nabla \cdot \left(\overline{\rho_{a}v''\Psi_{3,l}''}\right) - \frac{\partial}{\partial z}\left(v_{\text{sed},3,l}\bar{\rho_{a}}\hat{\Psi}_{3,l}\right) - Wa_{3,l} - Ca_{3,l} - Nu_{3,l} - Em_{3,l} - Co_{3,l} - Ch_{3,l} - Eq_{3,l}\right)$$

$$(13)$$

**13. Line 515: ARI is already defined earlier in text.**

Definition is removed

**14. The sub-labels in Figure 9 are difficult to read due to the white font color. Consider enclosing the letters in a contrasting box or background to improve visibility.**

**Corrected**

Figure 9. Comparison of net shortwave radiative flux estimated using MieAI against those estimated using Look-up table (LUT) approach for a case study involving the La Soufrière volcanic eruption (denoted by the black triangle) event simulated using ICON-ART. Here, panel a) shows the net SW flux estimated using LUT, b) shows the same estimated using MieAI and c) shows the absolute difference between them. The volcanic plume is depicted in panel d) whereas panel e) shows the mixed mode aerosols within the plume. Panel f) zooms panel c) over the plume region.

**15. Line 697: Online Emission Module→ OEM**

Corrected

**16. Line 712-713: vegetation photosynthesis and respiration model (VPRM)?**

 As mentioned earlier, we have added a short description of how VPRM is used for the simulation of CO2 including references to the original VPRM publication and to the first publication using VPRM within ICON-ART.

**17. Appendix D and E do not seem cited or discussed in the text.**

 Appendix D is indeed cited in section 3.2. A citation to appendix E is added to section 2.1:

XML tags and namelist settings for OEM are described in Appendix E.

---

## Author Comment (AC4)

**Response to Reviewers**

**Manuscript: Hoshyaripour et al. (2025), GMD**

We sincerely thank both reviewers for their detailed and constructive comments, which helped us improve the manuscript.

Reviewer comments are presented in blue, our replies follow in black, followed by the corresponding changes in the revised manuscript in italic format.

**Reviewer 2:**

- 1. L18: "essential for improving predictions related to weather, renewable energy, climate change, air pollution..." consider changing to: "essential for improving predictions and understanding related to weather, renewable energy, climate change, air pollution..."
- Revised accordingly:

Therefore, accurately simulating atmospheric composition is essential for improving predictions and **understandings** related to weather, renewable energy, climate change, air pollution, and associated health impacts.

- 2. L59-60: "OEM enables efficient processing of emissions that are constant in time or changing only temporally, but not spatially" This is not clear to me (and may also bother other readers): if different sources are varying differently temporally, it means emissions overall are changing spatially too please clarify this.
- The reviewer is right that this is not sufficiently clear. The spatial patterns of the
  emissions of individual categories are constant, but indeed the patterns of the total
  emissions (sum over all categories) usually change with time because the temporal
  profiles of the individual categories are different. We changed the text as follows:

OEM enables efficient processing of emissions which can be represented by adding up individual source categories, with the emissions from each category being fixed in space but varying over time.

3. While the paper excels at describing what has been implemented, it could be strengthened by briefly showing why some of the new developments matter in a more quantitative sense. For example: Section 2.4 (Sea Salt): The new Grythe et al. (2014) parameterization is included to better represent SST dependence. A simple zonal-mean plot or a brief statement quantifying the typical change in sea salt emissions or burden in tropical regions compared to the older scheme would be highly illustrative. Section 4.4 (Subpollen Particles): The parameterization for SPP release is described. It would be beneficial to include a sentence stating the typical order-of-magnitude contribution of SPPs to total aerosol number concentration or CCN in relevant regions during pollen season, even if citing another study (Werchner et al., 2022?). Section 3.2 (Detailed

chemistry mechanisms): a brief comment on the typical computational cost increase when moving from a simplified chemistry scheme (like Linoz) to a full mechanism (like MOZART-T1) would provide valuable context for users planning simulations. A percentage increase in runtime, similar to that provided for LINOZ in Section 3.1.3, would be sufficient.

• For the sea-salt emissions, we agree that assessing the impact of the Grythe et al. (2014) parameterization would indeed be valuable. However, a meaningful detailed quantitative comparison requires a comprehensive analysis of the full emission—transport—deposition cycle, rather than an isolated examination of the SST dependence. In addition to the different formulations, the two parameterizations also differ in their particle-size distributions, which further complicates a direct and fair comparison. Such an in-depth evaluation is beyond the scope of the present paper, whose primary aim is to document the model developments and implementation. We therefore prefer to address this analysis in a dedicated follow-up study, which is already underway. We have expanded Section 2.4 with a short qualitative description of the key conceptual differences between the two parameterizations, focusing on their treatment of whitecap coverage, particle-size distribution, and the explicit SST dependence introduced in Grythe et al. (2014).

MMS and G14 sea-salt emission schemes differ not only in their whitecap formulations but also in their treatment of particle-size distributions and SST-dependent scaling (Grythe et al., 2014; Barthel et al., 2019; Li et al., 2024). Barthel et al. (2019) demonstrated that SST corrections can substantially reduce coarse-mode concentrations and may even have a larger impact than switching between source functions. They also found the strongest differences for particles larger than PM2.5, with SST effects further amplifying these differences. These insights highlight that the structural contrasts between MMS and G14 schemes, particularly the inclusion of SST dependence and the size-resolved flux formulation, can significantly influence emitted mass. While a quantitative evaluation is beyond the scope of this study, this context helps to clarify the expected behavior of the new G14 implementation.

 For the subpollen particle (SPP) parameterization, we have now added a brief statement summarizing the typical magnitude of SPP contributions to total aerosol number and CCN concentrations based on literature values (Werchner et al., 2022):

Werchner et al., (2022) reported for a case study that SPP concentrations (only used as INP, not as CCN) vary between  $10^2$  and  $10^6$  m-3, with a mean value of  $4x10^3$  m-3 (especially relevant in warmer levels fit for biological ice nucleation), while mean pollen concentration amount to  $3.4x10^3$  m-3.

• In Section 3.2, the following sentence was added to address to question about the typical computational cost increase for full chemistry:

A full-chemistry simulation with MOZART-T1 increases the total runtime by roughly a factor of 10 compared to an ICON simulation without ART (tested on an HPC system with AMD Rome nodes with two AMD Epyc 7742 64-core CPU sockets each), reflecting not only the

computational cost of the chemical mechanism but also the additional overhead from tracer transport, emissions, deposition processes, and model output.

- 4. Figure 6: The inner and outer circles are unclear (visually and also in terms of values). For the winter plot (left), the outer rings mostly match the surrounding areas on the contour map but for the summer plot (right) these outer rings are consistently of a lighter shade than the surrounding values in the map which suggests some issue with sampling - please double check. If I disregard the outer rings and only compare the inner circle values with the surrounding values on the simulated map, I see a better model-obs agreement. However, when comparing the inner circles with outer rings, it looks like the model is underestimating surface ozone in both winter and summer. This underestimation doesn't sit well with the broader context of basically all global and regional models overestimating Northern Hemispheric surface ozone (e.g., Young et al., 2013; 2018; Ansari et al., 2025; Nalam et al., 2025, Gao et al., 2025). I suggest that the authors make this figure simpler by only showing one solid circle representing only observed values, and include the overall mean bias, RMSE, and correlation coefficient r for both seasons somewhere in the figure and the text. Accordingly, the text that "the model accurately reproduces..." should be made more nuanced and discussed in the broader context of the aforementioned papers. The authors must also mention which emission inventory was used for these LAM simulations over Europe. The authors should discuss potential reasons for O3 underestimation.
- We adapted the Figure to only show the observations as solid circles including mean bias, RMSE and correlation coefficient. The underestimation noted by the reviewer was caused by missing biogenic emissions of certain compounds in the model. These emissions have now been included, and a new simulation has been performed. The updated results show a slight overestimation, rather than an underestimation, during the summer period.

**Figure 6.** Mean afternoon ground-level O3 mixing ratios from ICON-ART simulations for winter (JF, left) and summer (JJA, right) 2019. Filled circles indicate observations from EMEP monitoring stations. Elevated sites and stations with less than 75% valid data were excluded.

- 5. L432: "hats and overbars"?
- We added the following description:

The transport equations in ICON-ART are Hesselberg-averaged (indicated by a hat) meaning a variable  $\Psi$  can be decomposed into a barycentric mean with respect to the air density  $\rho_a$  and its fluctuations:

$$\widehat{\Psi} = \Psi - \Psi^{\prime\prime} = \frac{\rho_a \Psi}{\rho_a}$$

The bar over a variable indicates Reynolds-averaging. The prognostic equations for number density  $(\widehat{\Psi}_{0,l})$  and mass mixing ratio  $(\widehat{\Psi}_{3,l})$  are solved at every fast physics time step and are given by: ...

- 6. L491: "processes such as removal processes" to "processes such as removal Mechanisms"?
- Changed to mechanisms
- 7. L493: Describe the key aspects of this alternative method in a couple of sentences here, especially in relation to its computational efficiency.
- The following text is added to the paper:

In contrast to ISORROPIA-2, this approach bypasses full thermodynamic equilibrium calculations and instead uses an empirical hygroscopic growth formulation for sea salt. This greatly reduces computational cost while retaining the key impact of water uptake on particle mass and related aerosol processes.

- 8. L571: "implemented in other models": name those models here along with the Citations.
- Done:

(HadCM (Woodward, 2001), LMDz-INCA (Balkanski et al., 2007), WRF-Chem (Zhao et al., 2013), MONARCH (Klose et al., 2021)).

- L669 (or thereabouts): Also include a couple of sentences on the best practices of using this dusty cirrus parameterization for different (coarser, finer, or variable) grid resolutions. How does it perform across scales? Has this been tested? This could be discussed a bit.
- Following Seifert et al. (2023, ACP), we added a short discussion on the recommended use of the dusty-cirrus parameterization at different model resolutions. Seifert et al. show that the scheme is generally robust across a range of grid spacings, as long as the

underlying dust mass and number concentrations are physically consistent. At coarser resolutions, the parameterization captures large-scale cirrus occurrence and radiative effects reasonably well, while finer or convection-resolving grids benefit from the improved representation of vertical motions and aerosol gradients that influence heterogeneous freezing. Although a systematic resolution-sensitivity analysis was not part of the study, available tests indicate consistent behavior of the parameterization across scales. We added the following text to the revised manuscript summarizing these points:

Dusty-cirrus parameterization performs robustly from mesoscale model resolutions (~10–20 km) down to convection-resolving scales (~1–3 km), as long as dust mass and number concentrations are physically reasonable. Coarser grids capture the large-scale cirrus response, while finer or convection-resolving scales benefit from enhanced representation of vertical motions and aerosol gradients.

- 10. Figure 11: The average OLR value should be shown in enlarged font or ideally printed over the map, or the reader might miss it. The technical name of the simulation experiment is not needed on the figure.
- This figure is revised accordingly (in the revised version as Fig 13)

20220506,  $00\ UTC+12.00\ h$ , overpass time: 20220506,  $12:01-12:09\ UTC$

Figure 13. Comparison of global ICON-ART simulation for 12 UTC of 6 May 2022 with CERES Level 2 satellite data of outgoing longwave radiation at the top of atmosphere. Observations (left), ICON-ART without dusty cirrus parametrization (center), and ICON-ART with dusty cirrus parametrization (right).

- 11. Figure 12: This schematic could be improved: include additional boxes at the top showing input data (for both ICON and ART). Name some typical variables (winds, moisture, pressure; anthro emissions). Similarly, name some typical output variables from ICON and ART; add additional boxes if necessary. Use appropriate arrows along the lines to indicate the direction of control and sequence of execution and data flow. Aim to better depict the loops and subloops within the model time integration workflow. In the caption, consider changing "circles" to "loops".
- This figure is revised accordingly:

**Figure 14.** Schematic of the coupling of ICON–ART. The sequence in which processes of ICON are executed is illustrated by the blue boxes. Processes of ART are illustrated by the orange boxes. An orange frame around a blue box indicates, that the according code is part of the ICON tracer framework but ART tracers are treated inside this framework. The gray and black loops indicate the sequences of the time integration. Some examples of input and output fields are also show at the top and bottom of the loop.

- 12. L698: "since also diagnostic variables can be defined with it" to "since diagnostic variables can also be defined with it".
- Done